# High-Fidelity Scientific Simulation Surrogates via Adaptive Implicit Neural Representations

## Abstract

Effective surrogate models are critical for accelerating scientific simulations. Implicit neural representations (INRs) offer a compact and continuous framework for modeling spatially structured data, but they often struggle with complex scientific fields exhibiting localized, high-frequency variations. Recent approaches address this by introducing additional features along rigid geometric structures (*e.g.*, grids), but at the cost of flexibility and increased model size. In this paper, we propose a simple yet effective alternative: Feature-Adaptive INR (FA-INR). FA-INR leverages cross-attention to an augmented memory bank to learn flexible feature representations, enabling adaptive allocation of model capacity based on data characteristics, rather than rigid structural assumptions. To further improve scalability, we introduce a coordinate-guided mixture of experts (MoE) that enhances the specialization and efficiency of feature representations. Experiments on three large-scale ensemble simulation datasets show that FA-INR achieves state-of-the-art fidelity while significantly reducing model size, establishing a new trade-off frontier between accuracy and compactness for INR-based surrogates.

## 1 Introduction

Accurately simulating complex physical systems is vital for scientific discovery, but high-fidelity simulations are often prohibitively expensive to run at scale (Chartier et al., 2021; Schneider et al., 2024; Wang et al., 2025). Surrogate models offer a practical alternative by learning from simulation outputs or observational data to deliver fast and accurate predictions (He et al., 2019; Catalani et al., 2024; Luo et al., 2024; Serrano et al., 2023a; Shi et al., 2022a;b). This enables researchers to efficiently explore scientific phenomena, test hypotheses, and support decision-making. For example, climate scientists can use surrogate models to study how ocean temperatures respond to varying wind stress without repeatedly running costly simulations.

Implicit neural representations (INRs) provide a compact and continuous framework for surrogate modeling, particularly well-suited to spatially structured data. INRs typically use neural networks—most commonly multilayer perceptrons (MLPs) (Catalani et al., 2024; Han & Wang, 2022; Mildenhall et al., 2021; Serrano et al., 2023a; Sitzmann et al., 2020)—to learn a mapping from input coordinates (*e.g.*, spatial locations in the ocean) and simulation conditions (*e.g.*, wind stress amplitudes) directly to target variables (*e.g.*, temperature). This formulation enables *resolution-independent*, continuous predictions *at arbitrary query points* with a compact model size, making INRs an appealing choice for accelerating simulations.

Despite these advantages, applications of INRs in complex scientific domains often suffer from limited fidelity due to *spectral bias* (Rahaman et al., 2019; Fridovich-Keil et al., 2022a)—particularly when modeling localized, high-frequency variations such as discontinuities or fine-scale structures that arise from underlying physical processes or complex boundary conditions (Brandstetter et al., 2022; Lu et al., 2021; Sitzmann et al., 2020). To address this limitation, recent work augments INRs with learnable embeddings defined over rigid geometric structures (*e.g.*, grids) to explicitly encode complex spatial variations (Cao & Johnson, 2023; Chen et al., 2025; Fridovich-Keil et al., 2023; 2022b; Sun et al., 2022a;b). In this setting, inputs to INRs are transformed into interpolated features retrieved from these embeddings. However, this approach compromises one of INRs' key

strengths: *compact modeling*. Specifically, feature grids introduce significant memory overhead and scale poorly with increasing dimensionality and spatial resolution, even when low-rank approximations are used (Cao & Johnson, 2023; Chen et al., 2022a; Fridovich-Keil et al., 2023). Moreover, data-independent rigid structures *fail to adapt to data characteristics,* often allocating excessive capacity to smooth regions while under-representing rapidly changing areas.

*How can we preserve INRs' compactness while improving their fidelity in modeling scientific data?*

Our proposal is to relax the rigid structural assumption. Specifically, unlike previous approaches, we bypass pre-defined positional structures (*e.g.*, grids) when augmenting INRs with learnable embeddings, enabling the model to allocate its capacity adaptively based on the data. We realize this idea using well-established cross-attention mechanisms (Vaswani et al., 2017) applied to a learnable, augmented key-value memory bank, inspired by the memory tokens in language modeling (Lample et al., 2019; Wu et al., 2022; Zhang & Cai, 2022). In this framework, grid-based embeddings emerge as a special case in which both the keys and interpolation weights are fixed a priori. By relaxing these constraints—and instead learning the keys alongside the values—our approach, **Feature-Adaptive INRs (FA-INR)**, unleashes the representational capacity of augmented embeddings. This flexibility directly addresses our central research question by enabling high-fidelity surrogate modeling while using significantly more compact embedding structures, as illustrated in Figure 1.

To promote more effective use of the key-value pairs in the augmented memory bank—motivated by our observation that some pairs were rarely utilized—we propose specializing them for distinct spatial locations. We implement this idea using the Mixture of Experts (MoE) framework (Chen et al., 2022c; Du et al., 2022; Fedus et al., 2022; Jacobs et al., 1991; Riquelme et al., 2021; Shazeer et al., 2017), where the augmented key-value pairs are partitioned into multiple expert groups, and a routing mechanism is defined based on spatial coordinates. Given an input coordinate and simulation condition, our INR model first routes it to specific experts (*i.e.*, memory banks), then uses the retrieved value embedding as input to an MLP to predict the target variable. This design allows each expert to focus on dedicated subregions of the task space, enhancing feature extraction and improving embedding efficiency—particularly in modeling complex scientific data.

We validate our FA-INR on three large-scale ensemble simulation benchmarks across diverse domains: oceanography (Larios et al., 2023; Ringler et al., 2013), cosmology (Almgren et al., 2013), and fluid dynamics (Mallinson et al., 2013). We hold out a subset of simulation conditions as the test set and use the remainder for training and validation. Both quantitative and qualitative evaluations consistently show that our approach outperforms existing methods, achieving high-fidelity surrogate modeling with significantly smaller model sizes.

Our **key contributions** include:

- Proposing FA-INR, an adaptive embedding-augmented INR that achieves high-fidelity surrogates with a compact model size.
- Establishing a novel connection between cross-attention over augmented memory and grid-based embedding augmentations, enabling a simple yet powerful implementation.
- Introducing a spatially-specialized memory mechanism based on MoE, which allows FA-INR to scale effectively to complex and large-scale simulation datasets. Additionally, we present a novel strategy for incorporating simulation conditions (*e.g.*, stress amplitudes), explicitly modeling the dependence of the target variable (*e.g.*, temperature) on the input simulation parameters.

**Remark.** Our novelty lies in a fresh perspective on the limitations of INR, coupled with a tailored solution that strategically integrates and adapts established deep learning components—such as cross-attention and mixture-of-experts (MoE)—to address them. We argue that such an approach offers greater impact and inspiration for the community than proposing yet another standalone method.

## 2 RELATED WORK

**Implicit Neural Representations.** INRs have emerged as a powerful method for learning continuous, memory-efficient representations of complex signals. A common approach employs multilayer perceptrons (MLPs) as the backbone, mapping input coordinates directly to signal attributes (Mildenhall et al., 2021; Saragadam et al., 2023; Sitzmann et al., 2020; Tancik et al., 2021;

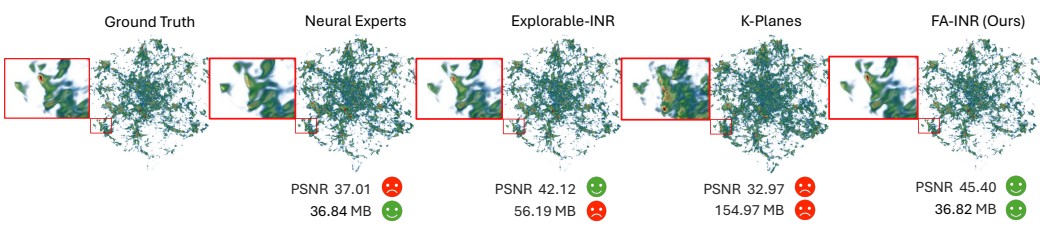

Figure 1: FA-INR outperforms alternative INR methods, including Neural Experts (a fully MLP-based INR) (Ben-Shabat et al., 2024), K-Planes (with a set of planes) (Fridovich-Keil et al., 2023), and Explorable-INR (with one grid and multiple planes) (Chen et al., 2025), while maintaining a compact model size.

2020). However, fully implicit MLP-based models suffer from a well-known spectral bias toward low-frequency components, making it difficult to capture high-frequency details such as edges and textures (Rahaman et al., 2019). While techniques such as positional encodings and specialized activation functions (Sitzmann et al., 2020; Mildenhall et al., 2021) mitigate this limitation, an alternative direction integrates explicit data structures into the implicit framework (Cao & Johnson, 2023; Fridovich-Keil et al., 2023; 2022b; Müller et al., 2022; Sun et al., 2022a;b). These hybrid approaches substantially enhance the reconstruction quality, though at the expense of increased memory for explicit representations.

Our approach addresses the above issue by a careful integration of cross-attention and Mixture-of-Experts (MoE), allowing the model to allocate the augmented memory efficiently and use it effectively. This sharply contrasts with other MoE usages in INR, which primarily focus on scalability (Di Sario et al., 2024; Ben-Shabat et al., 2024).

**Surrogate Models for Scientific Simulations.** Exploring the physical phenomenon across varying simulation conditions is often computationally expensive (Chartier et al., 2021; Ohana et al., 2024). To mitigate this, numerous deep learning (DL)-based surrogate models have been proposed to accelerate simulations at significantly reduced cost while preserving fidelity. For example, several studies focus on spatio-temporal forecasting (McCabe et al., 2024; Rühling Cachay et al., 2023; Lienen & Günnemann, 2022; Takamoto et al., 2023; Li et al., 2020), learning to approximate dynamic systems across both time and space. For mesh-based simulations defined on irregular, unstructured grids, graph- or mesh-aware models such as MeshGraphNets (Pfaff et al., 2020) and GNN-Surrogate (Shi et al., 2022b) have been developed. Some other works employed models like autoencoders for surrogate modeling of volumetric ensemble simulations (Shi et al., 2022a; Shen et al., 2024). However, most approaches are tied to discretized input spaces and designed to predict the entire discretized fields as outputs. As a result, they cannot support sparse querying and face scalability limitations due to high memory consumption. Furthermore, their architectures are often tailored for specific simulation types or domains, limiting their adaptability to broader scientific tasks.

In addition to these grid- and mesh-based surrogate models, Neural Operators (NOs) have emerged as another line of surrogate models that learn mappings between function spaces. Classical NOs such as DeepONet (Lu et al., 2019) and the Fourier Neural Operator (FNO) (Li et al., 2020) approximate the underlying PDE solution operator by taking entire input functions, such as initial conditions or boundary conditions, and predicting the corresponding solution fields. Recent work has extended NOs with transformer-based architectures, such as OFormer (Li et al., 2022), GNOT (Hao et al., 2023), GAOT (Wen et al., 2025), ViTO (Ovadia et al., 2024), and CViT (Wang et al., 2024), to better handle irregular geometries and multi-scale problems through self-attention and cross-attention mechanisms. Despite their expressiveness, these methods still rely on fully discretized representations of both input and output functions, resulting in significant memory and computational cost, particularly for high-resolution 3D domains. Thus, such models may not be well-suited for dense 3D ensemble surrogate modeling or applications that require sparse coordinate-based querying at arbitrary resolutions.

Implicit Neural Representations (INRs) take coordinates as inputs and directly learn continuous mappings to the output signals, providing a compact and efficient framework for various scientific applications, including simulation surrogates (Serrano et al., 2023b; Catalani et al., 2024; Chen et al., 2025), forecasting (Yin et al., 2022; Luo et al., 2024), and reduced-order modeling of PDEs (Chen et al., 2022b). Recent INR-based surrogate models have demonstrated strong performance in diverse

scientific domains, ranging from fluid dynamics and climate simulations (Pan et al., 2023; Luo et al., 2024) to medical imaging (Shen et al., 2022; Song et al., 2023; Reed et al., 2021). By learning grid- and mesh-agnostic representations, INRs *eliminate discretization constraints, reduce memory requirements, and enable flexible querying of continuous values at arbitrary coordinates or physical conditions*. Furthermore, INRs incorporated with advanced architectural designs are particularly effective in capturing fine-scale and high-frequency structures, and have been shown to outperform domain-specific surrogate models in applications such as ocean simulation (Chen et al., 2025).

**Remark.** Our work is specifically focused on *advancing INRs for high-fidelity scientific simulation surrogates*, particularly on large-scale ensemble datasets with 3D spatial coordinates and multivariate parameter conditions. In other words, the primary objective of our paper is not to develop the optimal surrogate modeling approach in a broad sense—in our opinion, given the diversity of scientific fields, there is hardly a one-size-fits-all approach. The advantages of INRs for surrogate modeling have been extensively investigated in prior studies (Pan et al., 2023; Chen et al., 2025; Shen et al., 2022; Song et al., 2023). Accordingly, we focus our comparisons on the most relevant and state-of-the-art INR-based methods to clearly demonstrate and validate our contributions.

## 3 INR-BASED SURROGATE MODELS

### 3.1 PRELIMINARIES

**Problem definition.** We aim to develop a surrogate to model spatially structured scientific data generated by a numerical simulator. This kind of simulator typically produces data corresponding to a predefined set of spatial coordinates, *i.e.*, $X = \{x_1, x_2, ..., x_N\}$, where each coordinate $x_i \in \mathbb{R}^d$. For given simulation parameters $p_j \in \mathbb{R}^m$ (*e.g.*, wind stress amplitudes), the simulator outputs a corresponding set of physical values $Y_j = \{y_{j,1}, y_{j,2}, ..., y_{j,N}\}$ at $X$'s locations. In our setting, *the initial condition is fixed*, and only the simulation parameters $p_j$ vary across ensemble members.

Typically, simulation involves solving complex partial differential equations (PDEs) (Ohana et al., 2024), easily requiring days to generate the data even for a small set of simulation parameters. The objective of a surrogate model is to learn an approximate mapping function $g : \mathbb{R}^{N \times d} \times \mathbb{R}^m \to \mathbb{R}^N$, such that given the coordinates $X$ and the simulation parameters $p_j$, it can predict the corresponding output field $Y_j$ significantly faster than the original simulators.

**INR for surrogate modeling.** An INR aims to learn this mapping using a neural network $f_\theta$, parameterized by weights $\theta$. Unlike traditional surrogate models that take the entire $X$ as input to predict $Y_j$ at once, an INR learns a function that maps a single input pair $(x_i, p_j)$ to its target output $y_{j,i}$. That is, $f_\theta : \mathbb{R}^d \times \mathbb{R}^m \to \mathbb{R}$, such that $f_\theta(x_i, p_j) \approx y_{j,i}$. Without loss of generality, we assume $y_{j,i}$ is a scalar value, but extending it to a vector is straightforward. Once the INR model is trained, it allows flexible querying at arbitrary coordinates and for arbitrary parameter sets.

Given a training dataset $\mathbb{D} = \{(X, p_1, Y_1), (X, p_2, Y_2), ..., (X, p_J, Y_J)\}$ with $J$ simulation runs, the INR model $f_\theta$ can be trained by minimizing the following empirical risk:

$$\mathcal{L}(\theta) = \frac{1}{J \cdot N} \sum_{j=1}^{J} \sum_{i=1}^{N} \ell(f_\theta(x_i, p_j), y_{j,i}), \tag{1}$$

where $\ell(\cdot)$ is the loss function, e.g., Mean Squared Error (MSE), which measures the differences between the predicted value $f_\theta(x_i, p_j)$ and the truth value $y_{j,i}$.

### 3.2 CHALLENGES

**Limitations of INRs.** While INR offers a compact framework for function approximation, directly processing raw coordinates as inputs often limits its ability to represent high-frequency variations (Fridovich-Keil et al., 2022a; Rahaman et al., 2019). To address this, positional encoding techniques, such as Fourier mapping (Mildenhall et al., 2021), are introduced to enable MLPs to better capture fine-grained details, where the input coordinate $x_i$ is transformed to a higher-dimensional space via $\gamma(x_i) = [\cos(2\pi\Omega x_i), \sin(2\pi\Omega x_i)]$, with $\Omega$ denoting the frequency matrix.

**Explicit representations for INRs.** Recent approaches further improve the representational power of INRs by incorporating *explicit* feature encodings into the MLP-based framework (Cao & Johnson, 2023; Fridovich-Keil et al., 2023; 2022b; Sun et al., 2022a;b), as illustrated in Figure 2. These encodings often employ structured representations such as feature grids, which store learnable feature vectors at discrete locations (vertices). For example, for 3D input coordinates, a feature grid $\mathcal{F} \in \mathbb{R}^{N_1 \times N_2 \times N_3 \times D}$ can be utilized. Here, $(N_1, N_2, N_3)$ defines the grid resolution, and $D$ is the dimension of the feature vectors stored at each vertex. To obtain the feature vector $z^{(x)}$ of an input coordinate $x \in \mathbb{R}^d$, the coordinate $x$ is first mapped from its original spatial domain to a normalized feature space $[-1, 1]^3$ through a linear projection $\phi$. Then, $z^{(x)}$ is retrieved by interpolating the features stored at the neighboring vertices of $x$ within the grid $\mathcal{F}$. The feature interpolation process can be defined as:



Figure 2: Architectures of three types of INR models: (a) Basic MLP-based INR, (b) INR with structured explicit representations, and (c) Our proposed Feature-Adaptive INR.

$$z^{(x)} = \psi(\mathcal{F}, \phi(x)). \tag{2}$$

While structured feature encodings such as grids and planes efficiently capture localized fine-grained details, achieving high-fidelity reconstructions typically requires dense representations, resulting in significant memory usage. This dense storage is often redundant, particularly for large-scale scientific datasets where smooth regions may present nearly uniform values. While in rapidly changing areas, these rigid data-independent structures may under-represent the underlying complexity.

## 4 FEATURE-ADAPTIVE INR

To address the limitations of previous approaches, we proposed an adaptive INR with an augmented *key-value memory bank*, inspired by similar concepts in language modeling. The memory bank can be denoted by $\{K \in \mathbb{R}^{M \times D_k}, V \in \mathbb{R}^{M \times D_v}\}$, where $K$ is a set of key vectors serving as feature locations, and $V$ is the corresponding set of feature vectors. By enabling both $K$ and $V$ to be learnable, thereby allowing the feature locations to be adaptive based on data characteristics rather than predefined locations, our approach makes more efficient and effective use of the augmented embeddings.

### 4.1 ADAPTIVE ENCODING VIA CROSS-ATTENTION

To retrieve the feature representation from the augmented memory bank, we leverage the cross-attention mechanism. As illustrated in Figure 3-(a), each query vector $q^{(x_i)}$ is produced from a spatial coordinate $x_i$ via an encoder MLP $f_{\theta_E}$, parameterized by $\theta_E$. Specifically, $f_{\theta_E}$ first maps $x_i$ to an intermediate feature vector $z^{(x_i)}$. This vector is then projected into a query representation via a linear mapping. Formally, this encoding process is defined as:

$$q^{(x_i)} = z^{(x_i)} W_q, \ W_q \in \mathbb{R}^{D_q \times D_k}, \qquad \text{where } z^{(x_i)} = f_{\theta_E}(x_i). \tag{3}$$

$\theta_E$ is the weights of the encoder MLP, and $W_q$ is a learnable projection matrix mapping the intermediate spatial features to the query dimension $D_k$.

Each spatial query dynamically retrieves the corresponding feature from the memory bank via cross-attention, similar to the interpolation process defined in Equation 2, but in a learnable and data-adaptive manner. The attention-based feature interpolation is computed as:

$$z^{(x_i, p_j)} = \text{Softmax}\left(\frac{q^{(x_i)}(KW_k)^\top}{\sqrt{D_k}}\right)V^{(p_j)}W_v, \tag{4}$$

where $W_k \in \mathbb{R}^{D_k \times D_k}$ and $W_v \in \mathbb{R}^{D_v \times D_v}$ are learnable linear projection matrices applied to keys and values, respectively. In particular, $W_k$ serves as a linear projection following the design of standard attention mechanisms, where separate projection layers are learned to provide greater

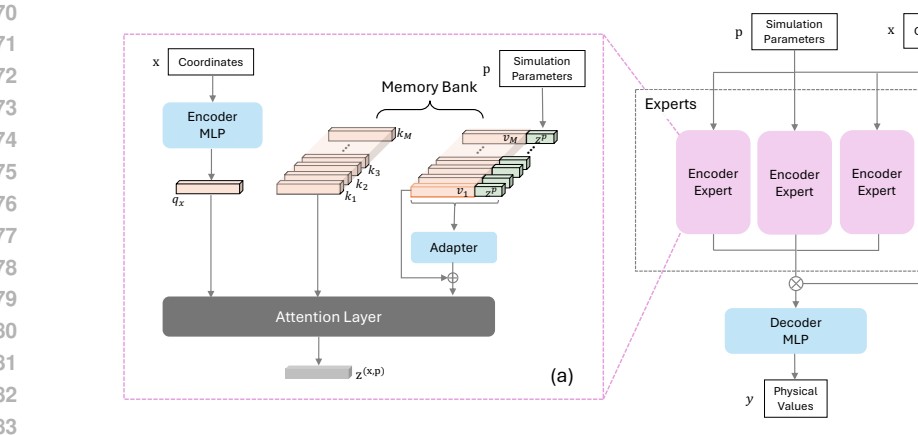

Figure 3: Architecture of the proposed FA-INR. An input pair $(x, p)$ is first routed by a gating mechanism in (b) to the most relevant spatially-specialized expert encoders (see (a)). Then, the aggregated feature vector from all selected experts is further processed by the MLP decoder in (b).

representational flexibility. $V^{(p_j)}$ represents the learned feature embeddings $V$ after conditioning on the simulation parameter $p_j \in \mathbb{R}^m$ (see subsection 3.1). See the next paragraph for details.

**Encoding simulation parameters.** We present a parameter-efficient approach to transform the learned embedding $V$ to $V^{(p_j)}$ based on the simulation parameters $p_j$, as illustrated in Figure 3. Given $p_j \in \mathbb{R}^m$, we first embed each dimension $s$ $(1 \le s \le m)$ separately into a feature vector to characterize each simulation variable. These $m$ embeddings are then combined via an element-wise (Hadamard) product to form an aggregated embedding $z^{(p_j)}$. As mentioned in (Fridovich-Keil et al., 2023), this embedding better preserves the signal from individual variables. This aggregated embedding $z^{(p_j)}$ is then concatenated with each of the $M$ learned embeddings in $V \in \mathbb{R}^{M \times D_v}$ (*i.e.*, the memory bank) and processed by an adapter $f_{\theta_A}$ (Houlsby et al., 2019) to generate $V^{(p_j)}$:

$$V^{(p_j)} = V + f_{\theta_A}(V \oplus z^{(p_j)}) \in \mathbb{R}^{M \times D_v}. \tag{5}$$

This residual connection is designed to better preserve the original information in $V$ while explicitly adapting the features based on the specific simulation parameters $p_j$.

## 4.2 SCALING UP FA-INR VIA MIXTURE-OF-EXPERTS (MoE)

As illustrated in Figure 4-(a), using a single small memory bank, our approach can already outperform the grid-based method. Intuitively, we would expect that enlarging the memory bank by adding more key-value pairs would achieve further improvements. However, as shown in Figure 4-(a), merely increasing the size of the memory bank results in limited performance gains or even slight degradation. With a closer inspection, we found out that many key-value pairs are rarely utilized. Specifically, we trace the Top-8 keys activated by each coordinate $x_i$ and summarize their usage in histograms. As shown in the bottom part of Figure 4-(b), a significant portion of keys are rarely used.

We hypothesize that this inefficient usage is caused by the random initialization of keys and values, which makes some pairs out of range to be retrieved. To address this issue, we propose to impose additional constraints to specialize key-value pairs into groups, and use the spatial location $x_i$ as the routing signal to determine which group of memory banks to query. This strategy just aligns with the idea of Mixture-of-Experts (MoE).

As illustrated in Figure 3-(b), each expert in our framework is an attention-based feature encoder associated with a memory bank. To dynamically select the most relevant experts for each coordinate $x_i$, we introduce a gating mechanism, consisting of two key components: a small, low-resolution feature grid $\mathcal{F}_G$ to provide a coarse spatial representation, and a small MLP $f_{\theta_G}$ parameterized by $\theta_G$. Given an input coordinate $x_i$, its corresponding coarse spatial feature $z_G^{(x_i)}$ is first extracted from $\mathcal{F}_G$. This feature vector is then processed by the small MLP $f_{\theta_G}$. Finally, a softmax function is applied to the output to produce the expert selection probabilities:

$$\Phi(x_i) = \text{softmax}(f_{\theta_G}(z_G^{(x_i)})), \ \Phi(x_i) \in \mathbb{R}^E. \tag{6}$$

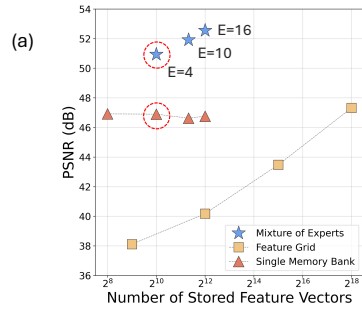 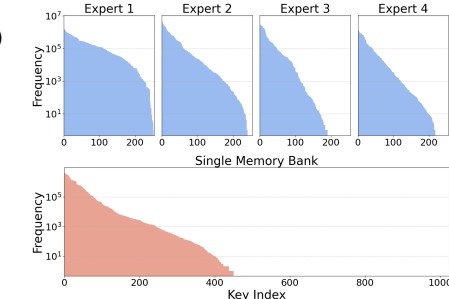

Figure 4: Comparison of scalability among three strategies on the MPAS-Ocean dataset. (a) Our FA-INR using MoE can effectively scale up the performance than solely expanding the memory bank size, while requiring fewer feature vectors compared to the grid-based method. The comparison of two histograms in (b) demonstrates that using MoE leads to more effective key utilization.

$E$ is the total number of expert modules; $\Phi(x_i)$ is the probability distribution over these experts.

### 4.3 OVERALL FRAMEWORK OF FA-INR

**Mixture of encoder experts.** Given the gating probabilities $\Phi(x_i)$, the Top-$K$ expert encoders will be selected for each coordinate $x_i$. The final aggregated feature $\bar{z}^{(x_i,p_j)}$ is then computed as a weighted sum of the feature vectors $z_k^{(x_i,p_j)}$ produced by the selected experts:

$$\bar{z}^{(x_i,p_j)} = \sum_{k=1}^{K} \Phi(x_i)_k \cdot z_k^{(x_i,p_j)}, \tag{7}$$

where $\Phi(x_i)_k$ is the probability of routing $x_i$ to the $k$-th expert, and $z_k^{(x_i,p_j)}$ represents the corresponding output feature vector from the $k$-th expert encoder, conditioned on both the spatial coordinate $x_i$ and the simulation parameter $p_j$.

**Feature decoder.** To decode the feature vector $\bar{z}^{(x_i,p_j)}$ into a physical scalar value, we employ a small MLP as a feature decoder, $f_{\theta_D}$, parameterized by $\theta_D$. This decoder maps the feature vector to the predicted physical value $\hat{y}_{j,i}$:

$$\hat{y}_{j,i} = f_{\theta_D}(\bar{z}^{(x_i,p_j)}). \tag{8}$$

**Optimization.** The entire FA-INR, including all encoder experts, the gating network, and the final feature decoder, is trained in an end-to-end manner. The optimization objective is to minimize the Mean Squared Error (MSE) loss between the predicted scalar values $\hat{y}_{j,i}$ and the ground-truth values $y_{j,i}$, computed across all $J$ ensemble members and $N$ coordinates. The training objective is formulated as follows:

$$\mathcal{L}_{\text{MSE}}(\theta) = \frac{1}{J \cdot N} \sum_{j=1}^{J} \sum_{i=1}^{N} \left\| f_{\theta_D}(\bar{z}^{(x_i,p_j)}) - y_{j,i} \right\|_2^2, \tag{9}$$

where $\theta$ represents all learnable parameters of the model, and $y_{j,i}$ is the ground-truth scalar value at coordinate $x_i$ for the $j$-th simulation run in the ensemble dataset.

## 5 EXPERIMENTS

### 5.1 EXPERIMENTAL SETUP

**Datasets.** We evaluate all models on three large-scale physics-grounded ensemble simulation benchmarks: one unstructured-mesh ocean simulation and two structured volumetric simulations. Specifically, *MPAS-Ocean* (Ringler et al., 2013) is a global ocean system simulation defined in spherical coordinates (*i.e.*, latitude, longitude, and depth) that models ocean temperature fields. The first structured volumetric dataset, *Nyx* (Almgren et al., 2013), is a cosmological hydrodynamics simulation with structured spatial grids and corresponding dark matter density measurements. The second, *CloverLeaf3D* (Mallinson et al., 2013), is a fluid dynamics simulation that captures energy values.

Table 2: Quantitative comparison of the proposed FA-INR with baseline methods on the MPAS-Ocean dataset. Both overall metrics and performance on high-frequency (HF) regions are reported.

| Model | #Experts | #Params. | PSNR↑ | MD↓ | PSNR (HF)↑ | SSIM ↑ |
|-------|----------|----------|-------|-----|------------|--------|
| FA-INR (Ours) | 10 | 1.10M | **51.72**$_{\pm0.45}$ | **0.1569**$_{\pm0.010}$ | **48.74**$_{\pm0.35}$ | **0.9935** $_{\pm5e\text{-}4}$ |
| Explorable-INR | - | 7.38M | 49.49$_{\pm0.31}$ | 0.1843$_{\pm0.008}$ | 46.08$_{\pm0.31}$ | 0.9908 $_{\pm3e\text{-}4}$ |
| K-Planes | - | 43.17M | 44.77$_{\pm0.72}$ | **0.1549**$_{\pm0.008}$ | 43.15$_{\pm0.56}$ | 0.9891 $_{\pm9e\text{-}4}$ |
| Neural Experts | 10 | 1.10M | 41.01$_{\pm0.71}$ | 0.3349$_{\pm0.025}$ | 38.16$_{\pm0.34}$ | 0.9811 $_{\pm9e\text{-}4}$ |
| MMGN | - | 1.10M | 42.99$_{\pm0.08}$ | 0.2444$_{\pm0.006}$ | 40.01$_{\pm0.23}$ | 0.9855 $_{\pm2e\text{-}4}$ |

Additional dataset statistics, including the number of ensemble members, the number of spatial coordinates, and the number of simulation variables, are provided in Table 1. Further details about each dataset are presented in Appendix A.

Table 1: Dataset statistics and training setups.

| Dataset | Simulation Parameters | Spatial Coords | Train Fields | Test Fields |
|---------|-----------------------|----------------|--------------|-------------|
| MPAS-Ocean | 4 | 11,845,146 | 70 | 30 |
| Nyx | 3 | $512^3$ | 100 | 30 |
| CloverLeaf3D | 6 | $128^3$ | 500 | 100 |

**Baselines.** We compare FA-INR against four baseline approaches: (1) MMGN (Luo et al., 2024) is a state-of-the-art INR-based surrogate model that employs a context-aware indexing mechanism along with a set of multiplicative basis functions to reconstruct high-quality physical fields from sparse observations. (2) Explorable-INR (Chen et al., 2025), a baseline surrogate model, combines a 3D feature grid with multiple 2D planes to balance model efficiency and accuracy. (3) Neural Experts (Ben-Shabat et al., 2024) enhances the capability of MLP-based SIREN (Sitzmann et al., 2020) by incorporating a MoE architecture to learn local, piecewise continuous approximations. (4) K-Planes (Fridovich-Keil et al., 2023) represents arbitrary-dimensional data by factorizing it into multiple 2D feature planes, providing a flexible and efficient surrogate modeling approach.

**Evaluation metrics.** We evaluate the performance of surrogate models using three metrics. Peak Signal-to-Noise Ratio (PSNR) (Huynh-Thu & Ghanbari, 2008) quantifies the numerical fidelity of predicted scalar fields relative to the ground-truth simulation, where higher PSNR indicates more accurate approximations of the true signals. To capture the worst-case deviations, we employ the normalized maximum difference (MD) (Shi et al., 2022b; Chen et al., 2025) to estimate the relative error bound. Finally, to measure perceptual quality, we compute the Structural Similarity Index Measure (SSIM) (Wang et al., 2004). Both predicted and ground-truth 3D fields are rendered into 2D images using identical viewpoints and rendering settings.

**Implementation and training details.** We employ a consistent model configuration for MPAS-Ocean and CloverLeaf datasets, while slightly increasing the model capacity for Nyx. Specifically, we use 256 key-value pairs in the memory bank for MPAS-Ocean and CloverLeaf, and 1,024 pairs for Nyx. To ensure fair comparisons, we also scale up the models of Neural Experts and MMGN by increasing their network depth and hidden dimensions to match the capacity of our model. This adjustment is necessary as both methods were originally designed for relatively simpler scenarios, such as 2D data. Additionally, we keep the original configuration of Explorable-INR and control the dimensions of our spatial and simulation parameter embeddings to match those used in its model. For the MoE component in FA-INR, we follow the standard sparsely-gated MoE design (Shazeer et al., 2017) using Top-2 expert routing. Additional implementation details are provided in Appendix C.

## 5.2 RESULTS ON UNSTRUCTURED-MESH SIMULATIONS

Table 2 summarizes the performance of all methods in approximating the simulation outputs across *unseen simulation parameters* on the MPAS-Ocean dataset. Our proposed method, FA-INR, consistently outperforms all baseline approaches, achieving an average improvement of 7.16 PSNR. Moreover, FA-INR attains the lowest MD, demonstrating its ability to suppress large, critical prediction errors. This reliability is particularly important for accurate and efficient parameter-space exploration in scientific research. While Explorable-INR achieves competitive performance, it relies on a combination of one feature grid and three high-resolution planes, leading to a model with approximately 7× more parameters than our model.

**Performance on high-frequency regions.** To further evaluate the capability of FA-INR in modeling fine-scale structures with high-frequency variations, we applied a 3D Discrete Cosine Transform (DCT) to the simulation fields and extracted the Top-5 high-frequency spatial components. We evaluated the performance of FA-INR against the baselines within these components. The last column in

Table 3: Quantitative comparison between FA-INR and four baselines on the Nyx and CloverLeaf datasets.

| Dataset | Method | #Experts | #Params. | PSNR↑ | MD↓ | PSNR (HF)↑ | SSIM↑ |
|---|---|---|---|---|---|---|---|
| Nyx | FA-INR (Ours) | 8 | 9.65M | **44.70**$_{\pm0.14}$ | **0.0819**$_{\pm0.0014}$ | **42.95**$_{\pm0.10}$ | **0.9573**$_{\pm1.6e\text{-}3}$ |
| | Explorable-INR | - | 14.73M | 43.09$_{\pm0.45}$ | 0.1173$_{\pm0.0074}$ | 38.54$_{\pm0.40}$ | 0.9430$_{\pm6.4e\text{-}3}$ |
| | K-Planes | - | 40.63M | 35.32$_{\pm0.73}$ | 0.1562$_{\pm0.0049}$ | 32.98$_{\pm0.42}$ | 0.8948$_{\pm4.9e\text{-}3}$ |
| | Neural Experts | 8 | 9.65M | 38.17$_{\pm0.45}$ | 0.1717$_{\pm0.0031}$ | 38.06$_{\pm0.62}$ | 0.9242$_{\pm6.1e\text{-}3}$ |
| | MMGN | - | 9.65M | 39.77$_{\pm0.22}$ | 0.2056$_{\pm0.0044}$ | 35.12$_{\pm0.19}$ | 0.9258$_{\pm4.2e\text{-}3}$ |
| CloverLeaf3D | FA-INR (Ours) | 10 | 1.10M | **53.40**$_{\pm0.49}$ | **0.0516**$_{\pm0.0024}$ | **45.42**$_{\pm0.43}$ | **0.9679**$_{\pm0.83e\text{-}2}$ |
| | Explorable-INR | - | 7.38M | 48.90$_{\pm0.17}$ | 0.0557$_{\pm0.0006}$ | 41.95$_{\pm0.15}$ | 0.9466$_{\pm0.11e\text{-}2}$ |
| | K-Planes | - | 6.79M | 46.35$_{\pm0.47}$ | 0.0641$_{\pm0.0029}$ | 39.05$_{\pm0.61}$ | 0.9406$_{\pm0.65e\text{-}2}$ |
| | Neural Experts | 10 | 1.10M | 52.40$_{\pm0.93}$ | 0.0639$_{\pm0.0036}$ | 43.69$_{\pm0.86}$ | 0.9662$_{\pm1.20e\text{-}2}$ |
| | MMGN | - | 1.10M | 45.52$_{\pm0.01}$ | 0.0697$_{\pm0.0002}$ | 38.54$_{\pm0.03}$ | 0.9312$_{\pm0.03e\text{-}2}$ |

Table 2 ("HF") shows that FA-INR consistently achieves the highest reconstruction accuracy across these challenging regions. These results demonstrate that our architecture is particularly well-suited for modeling localized, high-frequency variations in complex scientific data.

We also evaluate the generalization capability of FA-INR to *unseen spatial locations* (*e.g.*, new sensor positions). The corresponding results are reported in Appendix G. Ablation study and additional analysis on computational cost are provided in Appendix E and Appendix F.

**Remark.** We emphasize that the primary goal of our work is not to improve computational efficiency during training and inference. Instead, our focus is on advancing the use of INRs for *high-fidelity scientific simulation surrogates*. Thus, the most important metric is PSNR. As discussed in subsection 3.2, improving fidelity requires additional representations, yet determining where to allocate them remains a critical challenge. Our proposed FA-INR achieves the highest PSNR through *adaptively allocating explicit representations*. Therefore, we report the total parameter count, alongside our superior PSNR, mainly to demonstrate the effectiveness of our proposed strategy.

## 5.3 RESULTS ON STRUCTURED VOLUMETRIC SIMULATIONS

Compared to mesh-based simulations, volumetric simulations such as Nyx often exhibit denser and more complex 3D features. Additionally, spatial patterns in these datasets can vary significantly across different parameter conditions, as observed in CloverLeaf dataset. These characteristics pose additional challenges for INR-based surrogate models, which require accurate modeling of fine-grained spatial details and parameter-dependent variations.

The quantitative results are presented in Table 3. Overall, our FA-INR consistently achieves the highest performance across both datasets, with average improvements of 5.61 PSNR on Nyx and 5.11 PSNR on CloverLeaf. While Explorable-INR performs reasonably well on Nyx, its PSNR drops significantly on CloverLeaf (48.92 PSNR), indicating its limited adaptability to ensemble data with highly variable spatial features. In contrast, Neural Experts (with MoE) achieves competitive performance on CloverLeaf (53.38 PSNR) but underperforms on Nyx, suggesting that employing MoE architecture alone is insufficient. The strong and consistent performance of FA-INR across both benchmarks highlights the effectiveness of our expert specialization and adaptive feature encoding, enabling efficient and high-fidelity modeling of complex volumetric simulations.

**Visual fidelity.** To demonstrate the high visual quality achieved by our proposed FA-INR, we report the SSIM scores in Table 3 and provide qualitative comparisons on representative test fields from both datasets, as shown in Figure 5. Overall, FA-INR consistently achieves the highest visual fidelity, while K-Planes exhibits the most noticeable visual degradation across both datasets. As shown in the zoomed-in views in Figure 5-(a), K-Planes introduces numerous artifacts. These issues could be attributed to two reasons. First, the cosmology data contains highly complex features in the 3D space, which can be challenging to approximate solely using 2D representations. Second, the use of multi-resolution data structures could further amplify those false features. Due to space limit, additional qualitative analyses are presented in Appendix D.

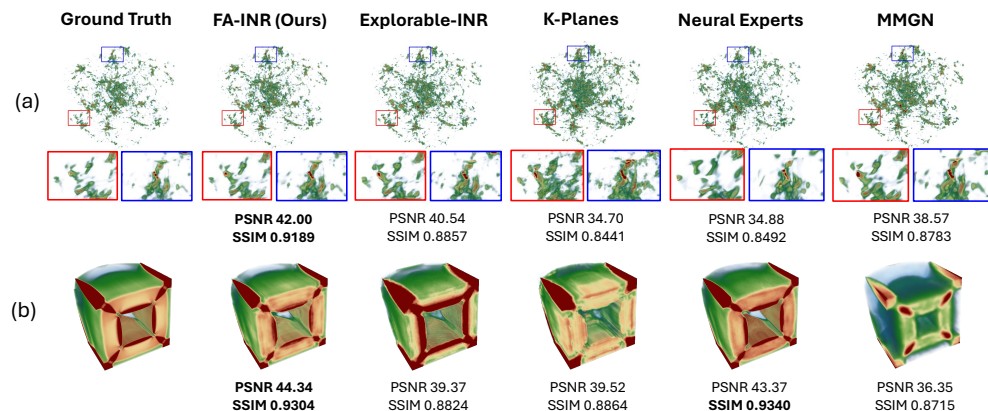

Figure 5: The volume rendering results on representative test fields from (a) Nyx and (b) CloverLead, across two volumetric ensemble datasets.

## 6 CONCLUSION

We introduce Feature-Adaptive INR (FA-INR), a compact, embedding-augmented implicit neural representation designed for high-fidelity surrogate modeling of scientific ensemble simulations. To overcome the rigidity of prior explicit data structures, we propose an adaptive encoding approach leveraging cross-attention with an augmented memory bank. For scalability to complex, large-scale ensemble datasets, we integrate a Mixture-of-Experts (MoE) framework that dynamically routes inputs to relevant memory banks based on their spatial data characteristics. Comprehensive evaluations have demonstrated that FA-INR achieves state-of-the-art fidelity surrogate modeling. Further discussion on limitations and future work is provided in Appendix H

## REPRODUCIBILITY STATEMENT

To ensure reproducibility of our work, we have included the experimental settings and implementation details in subsection 5.1 of the main paper and Appendix C. The source code of this work is available at an anonymous repository.

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

# APPENDIX

## A    EXPERIMENTAL DATASETS

In this section, we provide more details on the settings of simulation conditions.

**MPAS-Ocean** (Ringler et al., 2013) is a simulation dataset of the global ocean system. It is characterized by four key input parameters: Bulk Wind Stress Amplification (*BwsA*), Critical Bulk Richardson Number (*CbrN*), Gent-McWilliams eddy transport coefficient (*GM*), and Horizontal Viscosity (*HV*). This dataset comprises 100 simulation instances, each corresponding to a unique combination of these four simulation parameters. These are divided into 70 instances for training and 30 for testing.

**Nyx** (Almgren et al., 2013) is developed from the cosmological hydrodynamics simulations. This cosmology ensemble dataset was generated by using the following three simulation parameters: the total density of baryons (*OmB*), the total matter density (*OmM*), and the Hubble Constant (*h*). We use 100 instances for training and 30 instances for testing.

**CloverLeaf3D** (Mallinson et al., 2013) is a hydrodynamics simulation dataset generated by solving the 3D compressible Euler equations. This dataset focuses on six simulation parameters: three density parameters and three energy parameters. In the experiments, we randomly sample 500 instances for training and 100 instances for testing.

## B    TERMINOLOGY

In this section, we want to provide further explanations on the terms used in our paper. We define a "field" as the collection of physical values at predefined spatial locations, generated by a simulation run with a specific set of input simulation parameters. Within the scientific ensemble dataset, "ensemble instance" or "ensemble member" refers to a single such field.

In Table 1, terms like "training fields" or "test fields" represent the total number of ensemble instances sampled for training or testing. For example, "30 test fields from the Nyx dataset" denotes that 30 distinct fields were used for testing. Each of these test fields was generated using a unique set of simulation parameters. For the Nyx dataset, each simulation parameter set includes three variables, as detailed under the "simulation parameters" heading in Table 1.

## C    EXPERIMENTAL DETAILS

All models and experiments are implemented in PyTorch and conducted on NVIDIA A100 GPUs. For the MPAS-Ocean and CloverLeaf datasets, we use a one-hidden-layer encoder MLP with Sine activation (as shown in Figure 3-a), while for the Nyx dataset, a deeper encoder MLP with four hidden layers is employed. For all three datasets, the value adapter in the memory bank is implemented using a two-layer MLP. Additionally, all datasets share the same decoder MLP architecture (shown in Figure 3-b), consisting of three layers with ReLU activation. Other implementation details are introduced in subsection 5.1.

All models were optimized using the Adam optimizer and the mean squared error (MSE) as the training objective. During training, we randomly selected 25% of the input pairs consisting of spatial coordinates and simulation parameters (as defined in subsection 3.1) to be the validation set. The initial learning rate was set to $1 \times 10^{-4}$ with a decay rate of 10%. For Neural Experts, which utilize a fully MLP-based architecture, we employed a lower learning rate of $1 \times 10^{-6}$ to ensure training stability on the Nyx dataset. Moreover, all models are optimized using a consistent batch size for fair comparisons.

## D    QUALITATIVE RESULTS

In the main paper, we demonstrated the rendered images of the Nyx test dataset. In this section, we provide additional qualitative results for the MPAS-Ocean and CloverLeaf3D datasets. All images

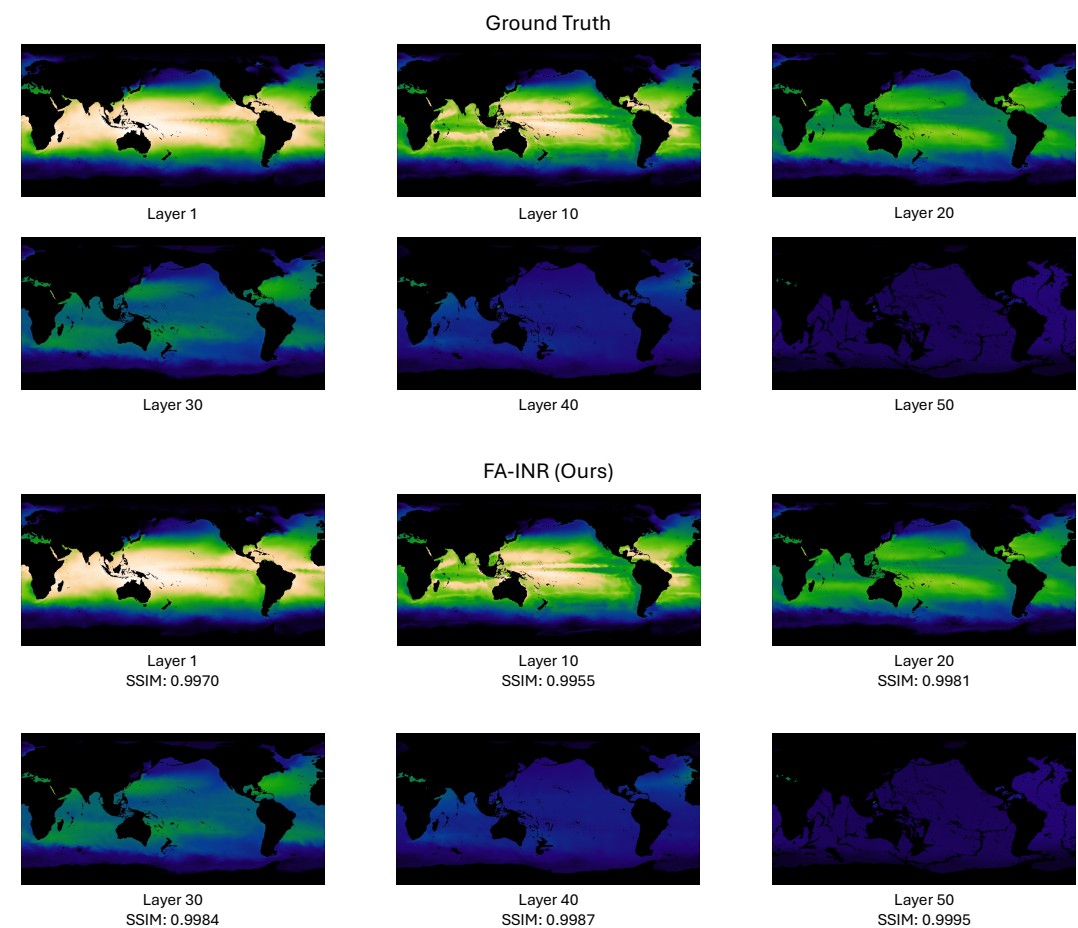

Figure 6: We select one test ensemble member from the MPAS-Ocean dataset, where each member contains 60 depth layers. In the visualizations, lighter colors indicate higher temperatures. Our proposed FA-INR consistently produces high-quality predictions across all depth levels.

were rendered using ParaView (Ayachit, 2015), an open-source platform built on the Visualization Toolkit (VTK) libraries.

## D.1 MPAS-OCEAN

Each instance in the MPAS-Ocean ensemble dataset comprises 60 depth levels. For visualization, we selected six layers from a single test instance, presented in Figure 6. We can observe that layers closer to the sea surface, such as layer 1 and layer 10, generally exhibit higher temperatures and contain more complex features.

Moreover, we compare the rendering results of our method with other baseline approaches as demonstrated in Figure 7 and Figure 8. A difference map is provided alongside each zoomed-in visualization to highlight prediction errors. All rendering results for the MPAS-Ocean dataset were generated using the visualization tool developed by GNN-Surrogate. [1] (Shi et al., 2022b)

## D.2 CLOVERLEAF3D

The visual comparisons between FA-INR and other baseline approaches are presented in Figure 9 and Figure 10, using two representative test instances from the CloverLeaf3D dataset. Compared

---

[1] https://github.com/trainsn/GNN-Surrogate

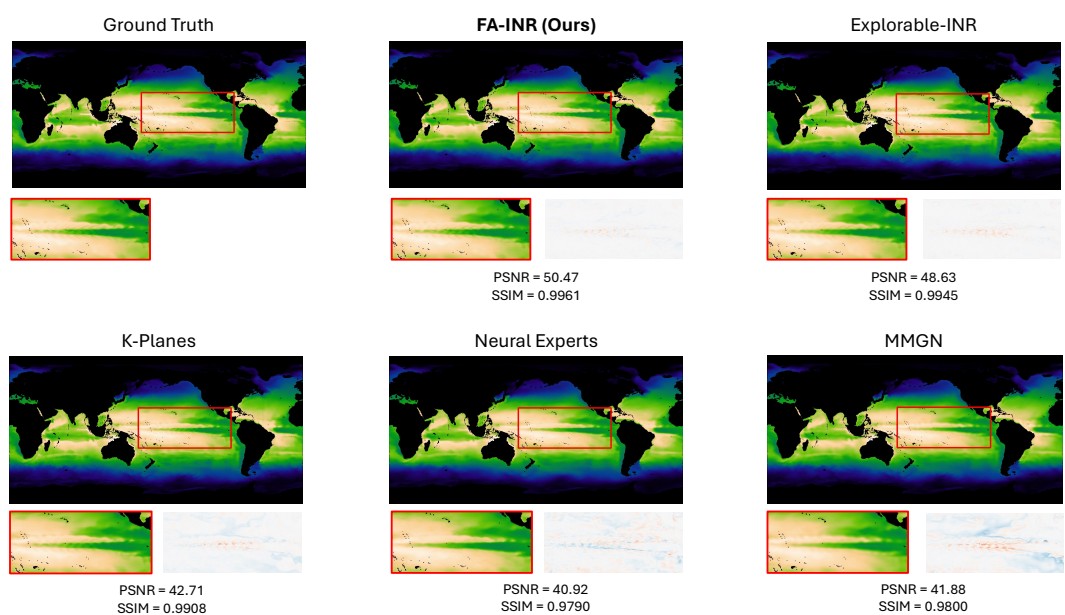

Figure 7: We select a layer close to the sea surface, which exhibits rich and high-variation features. Prediction errors are highlighted in the small figures on the right. The zoomed-in results reveal that the Neural Experts and MMGN struggle with modeling the high-frequency details.

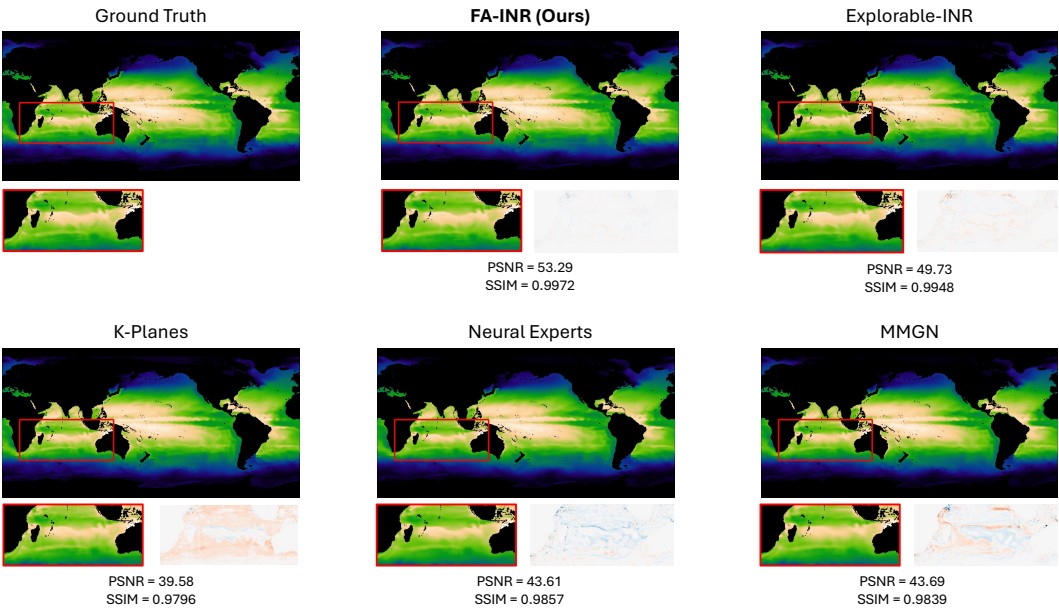

Figure 8: We select the same layer as in Figure 7 but from a different ensemble instance. Our proposed FA-INR achieves the highest prediction accuracy compared to the baseline approaches.

to the MPAS-Ocean results, larger visual differences are observed across all methods on this more challenging dataset. Both FA-INR and Neural Experts achieve top accuracy on these two cases. A possible explanation is that both methods utilize the Mixture-of-Experts (MoE) framework, which enables them to effectively handle the complex and high-variation patterns present in CloverLeaf3D.

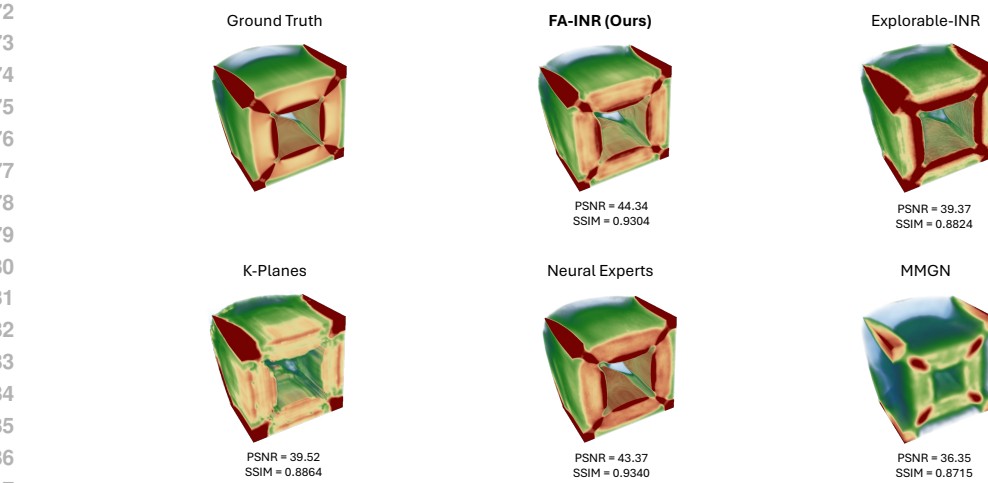

Figure 9: A representative test case from the CloverLeaf3D dataset is shown. Significant visual differences can be observed across the rendering results from all methods. Both FA-INR and Neural Experts, which utilize the MoE architecture, achieve the top prediction accuracy.

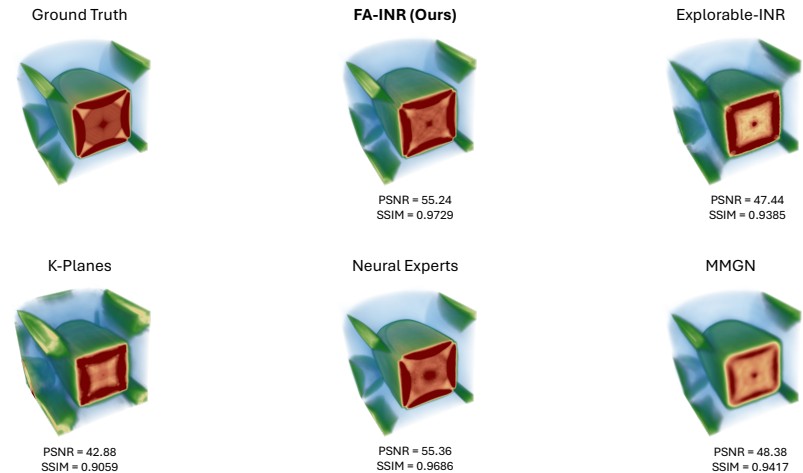

Figure 10: Another representative test case from the CloverLeaf3D dataset. Both FA-INR and Neural Experts, which utilize the MoE architecture, achieve the top prediction accuracy.

# E ADDITIONAL ABLATION STUDIES

## E.1 DESIGN CHOICES FOR SPATIAL ENCODING

To evaluate the impact of architectural choices for spatial encoding, we conduct ablation experiments on the Nyx dataset using different encoding configurations. Specifically, we compare our memory bank (*i.e.*, 1,024 key-value pairs per bank) with three alternative architectures: a learnable feature grid, a stack of feature planes, and an MLP with Sine activation. To ensure fair comparison, we control for total model size by maintaining approximately the same number of trainable parameters across all configurations. For all methods, we use a baseline embedding strategy where simulation parameters are embedded separately and integrated via concatenation with the spatial encoding before being passed to the decoder.

All results are summarized in Table 4. In addition to the baseline embedding strategy denoted as "w/o Adapter" in the table, we also include results from our original design incorporating the value

adapter module. Our key findings are as follows: (1) Our model with the adapter consistently outperforms all variants, both with and without MoE. (2) Feature grid and plane structures perform worse than MLP under the same parameter budget, suggesting their performance relies more heavily on dense spatial representations. (3) The adapter significantly improves performance over simple concatenation, especially when used with MoE.

| Design | Memory Bank | | Grid | Plane | MLP |
| | w/ Adapter | w/o Adapter | | | |
| --- | --- | --- | --- | --- | --- |
| w/o MoE | - | - | - | - | - |
| PSNR↑ | **39.49** | 38.98 | 36.69 | 33.41 | 38.19 |
| MD↓ | **0.0976** | 0.1281 | 0.1656 | 0.1896 | 0.1258 |
| #Experts | 8 | 8 | 8 | 8 | 8 |
| PSNR↑ | **44.72** | 44.05 | 41.10 | 38.77 | 42.77 |
| MD↓ | **0.0835** | 0.1028 | 0.1389 | 0.1408 | 0.1204 |

Table 4: Ablation study on alternative architectures for spatial encoding. Experiments are conducted on the Nyx dataset, both with and without the integration of a Mixture of Experts (MoE) framework.

## E.2 Design Choices for Parameter Encoding

In addition to the experiment where we replaced our proposed value adapter with the embedding concatenation, we evaluate the design of the parameter embedding mechanism used before attention. Specifically, we compare our proposed parameter-conditioned embedding mechanism (see subsection 4.1) against a simpler design where *parameter embeddings are concatenated with spatial embeddings before attention*.

This ablation is conducted on the MPAS-Ocean dataset. The simpler design achieves significantly lower performance of $47.79$ PSNR and $0.2014$ MD, in contrast to our proposed design, which achieves $51.92$ PSNR and $0.1536$ MD (as reported in Table 2). These results highlight the effectiveness of incorporating simulation parameters through parameter-conditioned attention, rather than relying on naive concatenation.

## E.3 Gating Module in FA-INR

As introduced in subsection 4.2, the gating module contains a low-resolution feature grid and a linear projection layer. In our main experiments, we fixed the feature grid resolution at $16^3$ across all three datasets, since using a small feature grid offers more efficient and stable optimization compared to other alternatives such as MLP, especially at the early training stage.

In this section, we investigate how varying the resolution of the feature grid impacts overall model performance. As shown in Table 5, a low-resolution grid ($16^3$) is sufficient to achieve high fidelity. Increasing the grid resolution does not necessarily lead to performance improvement, while the number of model parameters substantially increases. This observation is consistent with findings from previous studies (Ben-Shabat et al., 2024; Di Sario et al., 2024).

| Metric | $16^3$ | $32^3$ | $64^3$ |
| --- | --- | --- | --- |
| #Params | 1.10M | 1.56M | 5.23M |
| #Experts | 10 | 10 | 10 |
| PSNR↑ | 51.92 | 51.59 | 52.66 |
| MD↓ | 0.1536 | 0.1561 | 0.1605 |

Table 5: Results on varying the feature grid resolutions in the gating module on the MPAS-Ocean dataset.

## E.4 Alternative Approach for Expert Aggregation

The proposed FA-INR utilizes a Mixture of Experts (MoE) architecture with Top-2 routing strategy, effectively balancing training efficiency and prediction accuracy. This section discusses two alternative expert arrangement strategies.

First, instead of making the Top-2 selection, one alternative forwards each input to all expert encoders simultaneously. The outputs from all experts are concatenated into a single large feature vector, which is then projected back to the original feature dimension before being processed by the feature decoder. As presented in Table 6, the original MoE design with Top-2 expert selection outperforms this concatenation strategy, particularly achieving a 34% improvement in the MD metric. We hypothesize that the performance degradation arises because, unlike the MoE framework where experts can develop specialized knowledge, the concatenation method forces every expert to process all inputs. This can lead to duplicated representations across experts, substantially increasing training time while limiting prediction accuracy.

Second, we evaluated a *weighted sum* strategy over all experts, where the outputs from all experts are aggregated using learned weights instead of sparse gating. Although this approach provided a slight improvement in overall accuracy compared to Top-2 routing, as shown in Table 6, it substantially increased inference time. Specifically, on a single A100 GPU, the Top-2 routing strategy required only 1.70 seconds per test field, whereas the weighted sum strategy took 7.42 seconds. These results highlight that activating more experts per spatial input significantly increases computational overhead. Therefore, our sparsely-gated Top-2 routing approach offers a more practical balance between computational efficiency and model accuracy.

| Metric | MoE | Concatenation | Weighted Sum |
|---|---|---|---|
| #Experts | 10 | 10 | 10 |
| PSNR↑ | 51.92 | 50.75 | 53.51 |
| MD↓ | 0.1536 | 0.2061 | 0.1421 |

Table 6: Comparison of encoder expert utilization strategies (*i.e.*, Top-2 selection *vs.* all-expert concatenation *vs.* weighted sum over all experts) on the MPAS-Ocean dataset.

### E.5 NUMBER OF EXPERTS

We investigate the impact of increasing the number of encoding experts on model performance. In the experiments, all experts share the same architecture, and each coordinate is routed to the top-2 experts. As shown in Table 7, increasing the number of experts significantly improves performance on the MPAS-Ocean dataset. However, the performance gain becomes relatively small beyond using 10 experts, with a degradation in the MD score. A similar trend is observed for the Nyx dataset: increasing the expert number from 10 to 12 only provides a marginal improvement in PSNR, but with a decrease in the MD metric. It is important to note that the MoE framework in FA-INR is applied solely to the feature encoding component rather than the entire INR architecture. Therefore, increasing the number of experts will not substantially increase the training time and model size.

| Dataset | MPAS-Ocean | | | | Nyx | | | |
|---|---|---|---|---|---|---|---|---|
| #Experts | 1 | 4 | 10 | 16 | 1 | 8 | 10 | 12 |
| #Params. | 0.20M | 0.50M | 1.10M | 1.69M | 1.36M | 9.65M | 12.02M | 14.39M |
| PSNR↑ | 46.92 | 50.94 | 51.92 | **52.55** | 39.49 | 44.72 | 45.63 | **45.67** |
| MD↓ | 0.2096 | 0.1674 | **0.1536** | 0.1774 | 0.0976 | 0.0835 | **0.0715** | 0.0767 |

Table 7: Ablation study on the number of encoding experts.

## F COMPUTATIONAL EFFICIENCY AND MEMORY USAGE

The evaluation of computational cost and memory usage is summarized in Table 8. Compared to Explorable-INR, K-Planes requires an even larger model size. This is because K-Planes decomposes the input space into all possible pairs of variables and approximates each using a multi-resolution design, which substantially increases the number of required parameters. This challenge becomes particularly pronounced for ensemble simulations, where simulation parameters introduce additional input dimensions beyond those typically encountered in standard 3D/4D settings. In contrast, our proposed FA-INR not only achieves superior reconstruction quality but also utilizes significantly fewer model parameters.

Explorable-INR and K-Planes benefit from relatively fast training and inference times due to their simple, direct interpolation from feature grids or planes. While our FA-INR takes slightly longer runtimes, primarily because of cross-attention computations and the MoE gating mechanism, *these costs are valuable* as they lead to a much higher PSNR. Moreover, these additional times are considered to be negligible compared to running full-scale simulations (82.7 hours). We also note that, compared to other methods like Neural Experts and MMGN, our approach is more efficient in training and testing, while requiring significantly less training memory, even though efficiency is not our main focus.

| Method | PSNR (dB) | Training Time (hr) | Inference Time (sec/field) | Training Memory (M) | #Params. (M) | Model Size (MB) |
|--------|-----------|--------------------|-----------------------------|----------------------|--------------|-----------------|
| FA-INR (Ours) | 51.92 | 13.34 | 1.70 | 5966 | 1.10 | 4.19 |
| Explorable-INR | 49.45 | 8.54 | 0.33 | 1133 | 7.38 | 28.16 |
| K-Planes | 44.01 | 7.51 | 0.73 | 6731 | 43.17 | 164.68 |
| Neural Experts | 41.10 | 14.23 | 3.49 | 14883 | 1.10 | 4.42 |
| MMGN | 43.03 | 9.96 | 4.07 | 16698 | 1.10 | 4.40 |
| Simulation | – | 82.7 hr | | – | – | 903.7MB |

Table 8: Comparisons of training, inference efficiency, and computational memory usage across different surrogate models and traditional simulation.

# G  GENERALIZATION IN SPATIAL DOMAIN

## G.1  UNSTRUCTURED-MESH SIMULATIONS

While our proposed FA-INR surrogate is primarily designed to improve generalization across **unseen simulation parameters** (as demonstrated in subsection 5.2), we further evaluate its ability to generalize to **unseen spatial locations**. This evaluation provides complementary insight under a different generalization setting.

Specifically, during training, we randomly select 70% of the spatial coordinates within each of the 70 ensemble members and reserve the remaining 30% as a test set. This setting keeps the simulation parameters fixed and only evaluates generalization over spatial locations.

The quantitative results across all baselines on MPAS-Ocean dataset are summarized in Table 9 (*i.e.*, Trained columns). Overall, FA-INR consistently outperforms baseline INR models, achieving an average improvement of $5.86$ in PSNR and an average decrease of $0.1640$ in MD. Interestingly, Explorable-INR exhibits relatively poor spatial generalization compared to its performance in the main experiments. In contrast, K-Planes achieves competitive performance on both metrics, indicating that although K-Planes shows difficulty in generalization across simulation parameters, its multi-resolution feature planes are particularly effective for modeling spatial variations. However, this comes at the cost of significantly more model parameters.

We further evaluated all models on **unseen ensemble members**. This setting is more challenging as both spatial locations and simulation parameters were never observed during training. As shown in Table 9 (*i.e.*, Unseen columns), our method maintains the best performance among all INR baselines. K-Planes exhibits the largest degradation between the trained and unseen evaluations, indicating its limited generalization to new simulation parameters.

## G.2  STRUCTURED VOLUMETRIC SIMULATIONS

For structured volumetric simulations, we evaluate each model under a zero-shot super-resolution setting. During training, the data fields are downsampled from $128^3$ to $64^3$ for the CloverLeaf3D dataset and from $512^3$ to $256^3$ for the Nyx dataset, while evaluation is performed on the corresponding full-resolution fields. Performance on both the trained and unseen ensemble members is reported in Table 10 and Table 11, respectively.

On the CloverLeaf3D dataset (Table 10), our method outperforms all baselines on both the trained and unseen ensemble members. While MMGN achieves a slightly higher PSNR (55.25) on the trained fields, its performance drops substantially on unseen fields (45.41 PSNR), indicating its

| Metric | FA-INR (Ours) Trained | Unseen | Explorable-INR Trained | Unseen | K-Planes Trained | Unseen | Neural Experts Trained | Unseen | MMGN Trained | Unseen |
|---|---|---|---|---|---|---|---|---|---|---|
| #Params | 1.10M | | 7.38M | | 43.17M | | 1.10M | | 1.10M | |
| #Experts | 10 | | – | | – | | 10 | | – | |
| PSNR↑ | **52.20** | **51.43** | 49.89 | 49.15 | 51.04 | 43.38 | 40.28 | 39.60 | 44.17 | 39.27 |
| MD↓ | **0.1657** | **0.1736** | 0.3918 | 0.3918 | 0.1665 | 0.1730 | 0.4135 | 0.4046 | 0.3471 | 0.3534 |

Table 9: Evaluation of generalization to **unseen spatial locations** on the MPAS-Ocean dataset. We want to emphasize that *this spatial generalization task is different from the task in the main paper*, which focuses on generalization across unseen simulation parameters.

| Metric | FA-INR (Ours) Trained | Unseen | Explorable-INR Trained | Unseen | K-Planes Trained | Unseen | Neural Experts Trained | Unseen | MMGN Trained | Unseen |
|---|---|---|---|---|---|---|---|---|---|---|
| #Params | 1.10M | | 7.38M | | 6.79M | | 1.10M | | 1.10M | |
| #Experts | 10 | | – | | – | | 10 | | – | |
| PSNR↑ | **52.30** | **51.22** | 44.05 | 43.57 | 48.37 | 44.98 | 51.83 | 50.81 | **55.25** | 45.41 |
| MD↓ | 0.0459 | **0.0530** | 0.0609 | 0.0710 | 0.0538 | 0.0748 | 0.0532 | 0.0657 | **0.0388** | 0.0701 |

Table 10: Results of spatial domain generalization on the CloverLeaf3D dataset.

limited generalization. On the Nyx dataset (Table 11), our method again achieves the best results among all baselines for both trained and unseen ensemble members, demonstrating consistent high fidelity and robust generalization across scientific datasets.

## H  DISCUSSION

We propose Feature-Adaptive INR (FA-INR), a compact, embedding-augmented implicit neural representation designed for high-fidelity surrogate modeling of scientific ensemble simulations. Beyond its technical contributions, our approach also enables more interpretable predictions by allowing analysis of which model components contribute to outputs at specific spatial locations. This can be achieved by identifying the selected encoder experts and the most highly activated key-value pairs. Enhancing model transparency and reliability is particularly important for scientific applications.

**Limitation and future work.** We identify several directions for future work. First, we aim to expand our evaluation to include more ensemble datasets. However, our current setup, which covers three datasets from diverse scientific domains, already represents one of the most comprehensive evaluations compared to existing work on scientific simulations.

Second, one limitation is that FA-INR requires slightly longer training time compared to the INR models that rely on direct interpolation from the explicit structures (*e.g.*, Explorable-INR, and K-Planes). This is primarily due to our use of cross-attention mechanisms and the MoE framework. However, given the substantial time savings relative to running full numerical simulations, this additional training time is negligible in practice. For example, a single high-resolution Nyx run requires approximately 140 hours, and an MPAS-Ocean run requires about 83 hours. In the scenarios where hundreds or thousands of simulations are needed, the one-time training cost becomes minimal compared to the significant time saved through surrogate inference.

## I  EXPERT UTILIZATION

As discussed in subsection 4.2, the single-bank configuration presents inefficient key–value utilization, with many pairs *never* activated. Incorporating the Mixture-of-Experts (MoE) mitigates this issue and improves scalability, as demonstrated in Figure 4-(a). To further inspect expert usage dur-

| Metric | FA-INR (Ours) | | Explorable-INR | | K-Planes | | Neural Experts | | MMGN | |
|---|---|---|---|---|---|---|---|---|---|---|
| | Trained | Unseen | Trained | Unseen | Trained | Unseen | Trained | Unseen | Trained | Unseen |
| #Params | 9.65M | | 14.73M | | 40.63M | | 9.65M | | 9.65M | |
| #Experts | 8 | | – | | – | | 8 | | – | |
| PSNR↑ | 42.92 | **42.49** | 42.23 | 41.38 | 34.17 | 33.46 | 32.83 | 32.88 | 37.10 | 36.57 |
| MD↓ | 0.1030 | **0.1092** | 0.1259 | 0.1324 | 0.1808 | 0.1933 | 0.2874 | 0.2771 | 0.1589 | 0.1806 |

Table 11: Results of spatial domain generalization on the Nyx dataset.

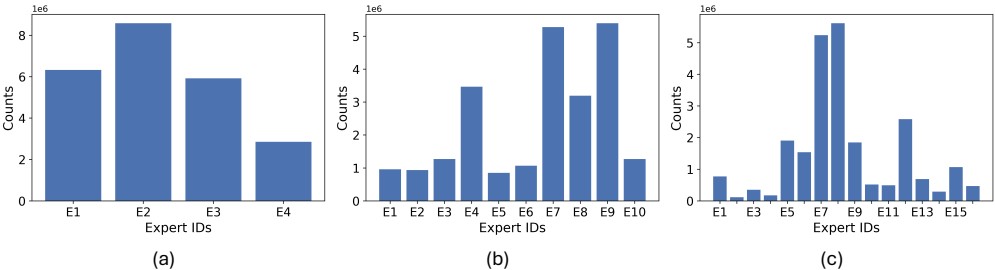

(a)    (b)    (c)

Figure 11: Visualization of expert utilization under three configurations with 4, 10, and 16 total experts (left to right). The 10-expert configuration (the middle) is the one adopted in the main FA-INR experiments.

ing inference, we count how many input coordinates are routed to each expert and summarize the results in Figure 11.

Across all configurations, **every expert is activated during inference**, although using an excessive number of experts (Figure 11-(c)) leads to more pronounced imbalance. In our main experiments, we adopt the 10-expert configuration (Figure 11-(b)), which offers a good trade-off between accuracy and effective expert utilization. The gating network naturally specializes experts based on data characteristics, resulting in uneven usage patterns that *indeed reflect the non-uniformity of the simulation data*. Similar behavior has been observed in prior INR methods that employ the MoE mechanism (Ben-Shabat et al., 2024), where certain experts are activated more frequently as some data features appear more often than others.

## J  THE USE OF LARGE LANGUAGE MODELS (LLMS)

LLMs were used solely for writing refinement and *did not* play any core role in the development of this work.

