# OpenReview forum: "High-Fidelity Scientific Simulation Surrogates via Adaptive Implicit Neural Representations"
_ICLR.cc/2026/Conference — Submitted to ICLR 2026_

### Official Review · Reviewer_yLiG · 2025-10-14

**Soundness:** 1
**Presentation:** 2
**Contribution:** 1
**Rating:** 2
**Confidence:** 4

**Summary:**

This paper proposes an implicit neural representation (INR)-based approach for fitting computationally simulated datasets. The method incorporates cross-attention-based adaptive feature augmentation and a mixture-of-experts strategy, achieving the highest peak signal-to-noise ratio (PSNR) among five models while using the same (or reduced) number of parameters. However, the title and overall presentation may be misleading, as the method aligns more closely with neural compression techniques than with computational simulation itself.

**Strengths:**

- The proposed method achieved an improved Pareto frontier between reconstruction quality and memory demand across the selected benchmark problems (though the benchmarks themselves are of limited broader significance; see Weaknesses).
- Section 4.2 provides a clear explanation of why further scaling up failed and justifies the use of a mixture of experts.
- Section 4 is presented with clear and detailed mathematical formulations.
- The inclusion of error bars in Table 2.
- The availability of code supports reproducibility.

**Weaknesses:**

1. The proposed method fits computationally simulated data by minimizing the empirical risk. However, this approach may not be directly aligned with surrogate modeling for computational simulations. In particular, lines 421--426 suggest that the primary goal of this work is data compression rather than simulation modeling.
2. Structural similarity index measure (SSIM) values are missing in Table 2, and Table 3 lacks error bars, maximum difference, and PSNR (high-frequency) metrics.
3. The heading of Section 4.2 appears somewhat inconsistent with lines 131–132. Additionally, the statement “they are well-suited for capturing fine-scale and high-frequency structures” (l.153) seems to contradict lines 122–128.
4. In Appendix G, varying the spatial grid across parameters is uncommon in ensemble simulations. Fixing “sensor locations” would typically be more representative of such settings.

**Questions:**

**Questions**

1. Could the authors clarify the meaning of the error bars in Table 2?
1. Could the authors clarify how the percentage improvements were computed (l.405, l.452–453)?
2. How does the attention map’s behavior differ for inputs $x$ in high- versus low-frequency regions?
5. Regarding the statement “This dense storage is often redundant” (l.238), could the authors clarify whether there exist methods to identify regions requiring dense storage *a priori*---that is, before obtaining the data?


**Suggestions for the Presentation**

1. The zoomed-in images in Figure 5a appear quite small and may benefit from enlargement for clearer comparison.
2. The discussion of INR limitations related to spectral bias appears multiple times in the main text; it might be helpful to consolidate these mentions.
3. The key research question (*How can we preserve INRs’ compactness while improving their fidelity in modeling scientific data?*) could be more clearly integrated with the rest of the narrative.
4. Since PSNR is already reported, the inclusion of MSE might be redundant.

---

> ### Author Response · Authors · 2025-11-23
> **Reply to Reviewer yLiG (1/2)**
>
> We sincerely appreciate your valuable time and effort in reviewing our paper. Thank you for the thoughtful and constructive feedback. Below, we would like to address each of the raised concerns in detail. We have already updated our paper with additional results, and the remaining revisions will be completed before December 3.
>
> > **Q: The proposed method fits computationally simulated data by minimizing the empirical risk. However, this approach may not be directly aligned with surrogate modeling for computational simulations. In particular, lines 421-426 suggest that the primary goal of this work is data compression rather than simulation modeling.**
>
> **A:**
> We appreciate the reviewer’s comment. We would like to clarify a potential misunderstanding regarding the objective of our work. Our work targets surrogate modeling rather than data compression.
>
> Our surrogate modeling framework is explicitly designed to learn an approximate function that maps input coordinates $X$ and simulation parameters $p_j$ to the corresponding physical values $Y_j$, significantly faster than the original simulator (now Lines 179-184). This mapping is learned directly from training data pairs via supervised learning, and is evaluated on the unseen simulation parameters to test the generalizability of the learned surrogate model (see Section 5.1 for the experimental setup). We mainly follow the existing work of data‑driven surrogate modeling, where the empirical risk minimization is the standard training paradigm. In other words, **our work is not about “data compression”**. In our humble opinion, compression means a model takes “Y” as input and outputs a compressed representation from which one can reproduce “Y.” *This sharply contrasts with our setting.*
>
> We also want to clarify Lines 450-451 (in the revised paper). When we said *“Thus, the most important metric is PSNR,”* it means we aim to improve the model’s performance on predicting the physical values of the unseen simulation parameters, which can be evaluated by PSNR between the true and predicted values.
>
>
> > **Q: Structural similarity index measure (SSIM) values are missing in Table 2, and Table 3 lacks error bars, maximum difference, and PSNR (high-frequency) metrics.**
>
> **A:**
> Thank you for the suggestion. We have **updated Tables 2 and 3 in the revised paper** to ensure that all evaluations are consistent. As shown in the updated tables, our model still outperforms the other baselines.
>
> > **Q: The heading of Section 4.2 appears somewhat inconsistent with lines 131-132.**
>
> **A:**
> We appreciate the reviewer’s careful reading. The concepts discussed in the introduction (now Lines 130-131) and the heading of Section 4.2 are **complementary rather than inconsistent**. Lines 130-131 motivate the need for improved scalability of FA-INR to handle the large, complex simulation datasets. Section 4.2 builds on this by showing that naively enlarging the memory bank cannot reliably improve performance, due to inefficient key utilization caused by random initialization. The introduction of the Mixture-of-Experts (MoE) mechanism **not only addresses scalability but also facilitates the key utilization** through spatially specialized experts. Thus, MoE serves two purposes: it (1) *enhances scalability* and (2) *mitigates an architectural limitation of key-value memory-based encodings*.
>
> > **Q: Additionally, the statement “they are well-suited for capturing fine-scale and high-frequency structures” (l.153) seems to contradict lines 122-128.**
>
> **A:**
> Thank you for raising this point. We respectfully believe that this perceived contradiction comes from a misunderstanding. Lines 122-128 (now Lines 121-127 in the revised paper) **refer specifically to basic fully MLP-based INRs**, which are known to struggle with high frequency unless additional mechanisms (e.g., positional encodings or explicit feature structures) are introduced. In contrast, the statement at Line 153 (now Line 164 in the revised paper) **refers to enhanced INR models** that incorporate these mechanisms, and which are the ones commonly adopted in recent INR-based surrogate modeling work. *We will update our paper to make this more explicit.*

---

> > ### Author Response · Authors · 2025-11-23
> > **Reply to Reviewer yLiG (2/2)**
> >
> > > **Q: In Appendix G, varying the spatial grid across parameters is uncommon in ensemble simulations. Fixing “sensor locations” would typically be more representative of such settings.**
> >
> > **A:**
> > Thank you for this comment. We would like to clarify that our additional experiments in Appendix G are **not** *intended to replicate a standard ensemble simulation setup*. Instead, they are used to demonstrate the spatial generalization capability of our INR-based surrogate model. While the ensemble simulations typically operate over fixed spatial locations, many downstream applications may require the surrogate models to generalize to unseen or shifted spatial locations. Given that INRs are resolution-independent and support continuous spatial queries, this evaluation is designed to evaluate the robustness of our model.
> >
> > > **Q: Could the authors clarify the meaning of the error bars in Table 2?**
> >
> > **A:**
> > Thank you for the question. The error bars in Table 2 indicate the standard deviation computed over five runs with different random seeds to reflect the robustness and stability of the model performance.
> >
> > > **Q: How does the attention map’s behavior differ for inputs in high- versus low-frequency regions?**
> >
> > **A:**
> > We appreciate the reviewer for this insightful question. Overall, we observed consistently *different attention behaviors between inputs from low-frequency (LF) and high-frequency (HF) regions*. **LF regions** correspond to smooth areas with low spatial variation. Their attention maps present a more concentrated distribution, where the top few keys receive much larger attention weights. These key-value pairs capture the dominant global structures learned consistently across the training fields, allowing the model to represent LF regions efficiently using only a relatively small set of basis features. In **HF regions**, the local field contains sharp variations. The corresponding attention weights are more distributed with lower magnitude, indicating that HF regions rely on a broader combination of keys to approximate the complex highly varied features. This distinction in attention behavior demonstrates that FA-INR adaptively allocates representational capacity based on the local characteristics of the field.
> >
> > > **Q: Regarding the statement “This dense storage is often redundant” (l.238), could the authors clarify whether there exist methods to identify regions requiring dense storage a priori---that is, before obtaining the data?**
> >
> > **A:**
> > We appreciate the reviewer’s thoughtful question. The statement *“This dense storage is often redundant”* (now Line 234) refers to the fact that methods based on fixed explicit structures must increase the resolution of those structures (e.g., grids, planes) to support high-resolution data. However, they allocate memory uniformly across the entire spatial domain, even in regions where the underlying field varies smoothly and does not require such high representational capacity.
> > Regarding the question *“identifying regions that require dense storage a priori”*: in general, **this is challenging for ensemble simulations**. The spatial complexity of the field depends on the simulation parameters, which vary across ensemble members. Thus, it is difficult to predetermine which regions will require larger representational capacity before seeing the data.
> >
> > This limitation motivates us to introduce the **learnable key-value locations** and the **MoE-based spatial routing**. Our model adaptively allocates representational capacity to complex regions in a data-driven manner, **without relying** on prior knowledge of where dense representation is needed.
> >
> > > **Q: The zoomed-in images in Figure 5a appear quite small and may benefit from enlargement for clearer comparison.**
> >
> > **A:**
> > Thank you for the helpful suggestion. We will enlarge the zoomed-in figures in Figure 5(a) to facilitate clearer comparisons.
> >
> > > **Q: The discussion of INR limitations related to spectral bias appears multiple times in the main text; it might be helpful to consolidate these mentions.**
> >
> > **A:**
> > Thank you for the suggestion. To improve clarity, we will revise our paper to consolidate the discussion of spectral bias.
> >
> > > **Q: The key research question (How can we preserve INRs’ compactness while improving their fidelity in modeling scientific data?) could be more clearly integrated with the rest of the narrative.**
> >
> > **A:**
> > Thank you for the suggestion. We will revise the introduction to better integrate this key research question with the rest of the paragraphs.

---

> > > ### Comment · Reviewer_yLiG · 2025-11-23
> > >
> > > Thank you very much for revising figures and tables, and also for making a promise for consolidation.
> > >
> > > Since the authors have pointed out that I misunderstood the theme of the paper, surrogate models for scientific simulations, I have decreased my confidence from 4 to 3.

---

> > > > ### Author Response · Authors · 2025-11-23
> > > > **Re: Official Comment by Reviewer yLiG**
> > > >
> > > > Dear reviewer yLiG,
> > > >
> > > > Thank you for your prompt reply. We are glad that you find our revised figures and tables useful.
> > > >
> > > > Regarding the theme, *surrogate models for scientific simulations,* our rebuttal was prepared to answer your question/concern, and we remain open for further discussion.
> > > >
> > > > Meanwhile, we were wondering whether our rebuttal addressed your concerns. The original rating is "Rating: 2: reject, not good enough," and we humbly believe that our rebuttal has addressed the items listed in weaknesses and questions. We would appreciate it if you could give us concrete feedback on our rebuttal, so we can further address the remaining questions. If our rebuttal has addressed most of your concerns and you feel satisfied with it, we would appreciate it if you could reconsider the rating.
> > > >
> > > > Best,
> > > > Authors

---

> > > > > ### Comment · Reviewer_yLiG · 2025-11-24
> > > > >
> > > > > Thank you for the detailed rebuttal. Unfortunately, my overall assessment remains unchanged. Framing the work as surrogate modeling for scientific simulation still feels too general. In particular, the paper does not address initial-value problems, which are central to many practical applications such as numerical weather prediction.
> > > > >
> > > > > From the computer vision perspective -- where I acknowledge my expertise is limited -- Reviewer Mqe9 has already pointed out that the technical novelty appears modest. From the scientific simulation perspective -- where I am more confident -- the absence of comparisons to neural operators is a significant gap, especially given their substantial track record in data-driven surrogates for PDE solvers.
> > > > >
> > > > > Additionally, although the paper claims to perform physical simulations, the mathematical setup is not adequately described. Key elements are missing: e.g., the parameter ranges, the explicit formulation of the 3D compressible Euler equations, and clarity about what is meant by “density” and “energy” being treated as parameters despite these quantities being part of the state variables solved for in the PDE. Without these details, it is difficult to interpret the experimental design as anything beyond dataset-driven curve fitting with sophisticated architecture.

---

> > > > > > ### Author Response · Authors · 2025-11-24
> > > > > >
> > > > > > Thank you for your prompt response. We will follow up with more details soon.

---

> > > > > > > ### Author Response · Authors · 2025-12-03
> > > > > > >
> > > > > > > We appreciate your follow-up comments. We note that these additional concerns went beyond the original review. Nevertheless, we try our best to address them as follows.
> > > > > > >
> > > > > > > First, we are fully aware of the breadth and diversity of scientific simulation, which spans many application domains. In this work, we **specifically focus on the problems where INRs have already demonstrated appropriateness and usefulness**, and our goal is to advance INR-based models for high-fidelity surrogate modeling. This focus is clearly articulated in our abstract, introduction (the second paragraph), and related work section (Lines 147–154 in the original submission). In other words, we never claim that our method is suitable for all scientific simulation surrogate scenarios, and we will be happy to adjust our title, if needed, to reflect it.
> > > > > > >
> > > > > > > We appreciate your comments regarding initial-value problems and PDE-based neural operators. While they are important to many scientific simulations, we would like to clarify that the term simulation **surrogate** is broad. In our humble opinion, *any method that accelerates simulation or provides a fast approximation of a simulation-generated field can be considered a surrogate*. Different domains emphasize different aspects. Some approaches aim to speed up PDE solvers, while others, such as INR-based methods, focus on directly **approximating the input-output relationship** of **high-resolution** scientific fields.
> > > > > > >
> > > > > > > Given the diversity across scientific simulation tasks, there is no single surrogate model that universally fits all settings. Neural operators have been widely adopted for learning solution operators of PDEs, whereas INRs have proven effective for learning continuous coordinate-based representations of complex fields. Our work builds on this latter line of research and specifically targets **accelerating parameter-space exploration**, which is **a key requirement in many ensemble simulation workflows**. We also refer the reviewer to our response to Reviewer Mqe9’s third question for additional clarification.
> > > > > > >
> > > > > > > We want to emphasize that we are not the first paper to use INRs for simulation surrogates. Prior works, such as [1–6], have already demonstrated their effectiveness in this setting. In the related work section, we clearly position our contributions within this line of research and contrast our approach with other surrogate models. Importantly, **the problem domains that we focus on are rooted in real scientific simulation workflows, where existing papers do not typically compare against neural operators**. The use of INR-based models as surrogates is already well established and broadly recognized by the community.
> > > > > > >
> > > > > > > In our humble opinion, what makes the research community exciting is its willingness to embrace new thinking or approaches, even if they do not explicitly model the underlying process. For example, in visual recognition, some approaches focus on modeling the image formation process (e.g., Bayesian or generative models), while others directly learn a mapping from images to labels. A similar contrast exists in scene reconstruction. Traditional graphics methods explicitly model scene geometry, materials, and light transport, whereas recent approaches, such as Gaussian splatting, bypass these physical formulations by using Gaussian primitives. We thus respectfully disagree with the newly raised concern that a surrogate model must *explicitly* address initial-value problems, Euler equations, or PDEs.
> > > > > > >
> > > > > > > Regarding the mathematical setup, we respectfully believe that these details fall outside the scope of our work. As clearly stated in our problem setup (Section 3.1), our goal is to learn and generalize the input-output relationship from simulation data. We would be happy to provide details of the parameter ranges, the explicit formulation underlying the simulations, and the definitions of density and energy. *These are clearly described in the papers associated with the simulation systems (see Appendix A for citations), and the datasets we use are well-known and widely adopted in the community.* However, we think they are orthogonal to our focus.
> > > > > > >
> > > > > > > Furthermore, we appreciate your feedback regarding technical novelty, and we have addressed related concerns in our response to Reviewer Mqe9.
> > > > > > >
> > > > > > > [1] Continuous Field Reconstruction from Sparse Observations with Implicit Neural Networks.
> > > > > > >
> > > > > > > [2] Explorable INR: An Implicit Neural Representation for Ensemble Simulation Enabling Efficient Spatial and Parameter Exploration.
> > > > > > >
> > > > > > > [3] INFINITY: Neural Field Modeling for Reynolds-Averaged Navier-Stokes Equations.
> > > > > > >
> > > > > > > [4] MC-INR: Efficient Encoding of Multivariate Scientific Simulation Data using Meta-Learning and Clustered Implicit Neural Representations.
> > > > > > >
> > > > > > > [5] Aero-Nef: Neural Fields for Rapid Aircraft Aerodynamics Simulations.
> > > > > > >
> > > > > > > [6] Neural Fields in Visual Computing and Beyond.

---

> > > > > > > > ### Author Response · Authors · 2025-12-03
> > > > > > > >
> > > > > > > > In addition, we have further updated our manuscript to incorporate the following revisions **(all revisions are highlighted in blue)**:
> > > > > > > >
> > > > > > > > 1. In the related work section, we have **added clarification** for our discussion regarding “INRs for capturing fine-scale and high-frequency structures” to avoid potential misunderstanding.
> > > > > > > >
> > > > > > > > 2. We have **consolidated our discussion regarding “spectral bias”** by revising both the introduction and related work sections to make this concept more explicit, reducing redundancy, and citing the relevant literature.
> > > > > > > >
> > > > > > > > 3. We have **revised our introduction section** to better integrate the research question, *“How can we preserve INRs’ compactness while improving their fidelity in modeling scientific data?”*, with the subsequent paragraphs.
> > > > > > > >
> > > > > > > > 4. We **have enlarged the zoomed-in views in Figure 5(a)** to facilitate clearer comparisons.

---

### Official Review · Reviewer_jjGs · 2025-10-27

**Soundness:** 3
**Presentation:** 4
**Contribution:** 3
**Rating:** 8
**Confidence:** 2

**Summary:**

The authors propose Feature-Adaptive INR (FA-INR), a method that uses cross-attention with an augmented memory bank to learn adaptive features for INRs in scientific simulation surrogate tasks, thus drastically reducing the surrogate model’s memory footprint. The authors also propose a mixture-of-experts architecture based on FA-INR that improves the scalability of their proposed method. The authors compare FA-INR against four other baselines across structured and unstructured grid problems.

**Strengths:**

The problem this paper seeks to solve is well-motivated (even for those not familiar with the state-of-the-art in INRs for accelerating scientific simulations), and the method itself is well-written and clear. I appreciate the depth and breadth of the experiments, which are performed across a variety of grid types and scientific domains.

**Weaknesses:**

Apart from the improvements in memory, one of the major benefits of using INRs is the fact that they directly model the output field and can thus be queried anywhere. In many scientific applications, it is possible that training data may arrive on different grids or sensor locations; INRs can deal with this directly. It is also possible that we may want to query the output of the INR at different locations than those seen during training.

I see in the appendix that the authors performed spatial domain generalization. How does FA-INR compare to the baselines at zero-shot super-resolution or zero-shot “grid transfer” (train on one grid and evaluate zero-shot on another) for the other datasets explored? How does FA-INR compare to the baselines at different sparsities of training spatial coordinates? I would recommend the authors explore a bit more fully these spatial domain generalization questions.

**Minor notes:**
I appreciate the idea of Figure 1, but it is a bit difficult to understand from first glance which method performs best without a corresponding ground truth figure.

**Questions:**

1. If $K$ is already a learnable set of keys, what is the purpose of $W_k$?
2. The authors mentioned that many key-value pairs are not heavily utilized. Do you also notice similar behavior with experts? Are only a few experts “activated” during inference, or is the coordinate-conditioning effective in activating them?
3. In this paper, I assume that the simulation parameters $p$ are finite-dimensional. Is it possible to also apply this method (with minor modifications) in the operator learning case when $p$ is an infinite-dimensional function?

---

> ### Author Response · Authors · 2025-11-23
> **Reply to Reviewer jjGs**
>
> We sincerely appreciate your valuable time and effort in reviewing this paper. We are glad that our paper was found to be “well-motivated,” “well-written,” and “clear,” and we thank the reviewer for recognizing the depth and breadth of our experiments. Below, we provide detailed responses to each comment. We have already updated our paper with additional results, and the remaining revisions will be completed before December 3.
>
> > **Q: I see in the appendix that the authors performed spatial domain generalization. How does FA-INR compare to the baselines at zero-shot super-resolution or zero-shot “grid transfer” (train on one grid and evaluate zero-shot on another) for the other datasets explored? How does FA-INR compare to the baselines at different sparsities of training spatial coordinates? I would recommend the authors explore a bit more fully these spatial domain generalization questions.**
>
> **A:**
> Thank you for the insightful suggestions. In the revised manuscript, we have **expanded our analysis of spatial domain generalization in Appendix G**. For MPAS-Ocean, we now report both spatial generalization within the trained ensemble members and generalization to unseen ensemble members (see Appendix G.1). For the structured volumetric datasets (Nyx and CloverLeaf3D), we further evaluate FA-INR under **a zero-shot super-resolution setting** and compare against all baselines on both trained and unseen ensemble members (see Appendix G.2). Across these settings, FA-INR consistently achieves the best performance. While MMGN achieves a slightly higher PSNR on the trained fields of the CloverLeaf3D dataset, its performance degrades substantially on unseen ensemble members, indicating much more limited generalization compared to our method.
>
> > **Q: Minor notes: I appreciate the idea of Figure 1, but it is a bit difficult to understand from first glance which method performs best without a corresponding ground truth figure.**
>
> **A:**
> Thank you for the suggestion. We will update Figure 1 in the revised version of our paper.
>
> > **Q: If $K$ is already a learnable set of keys, what is the purpose of $W_k$?**
>
> **A:**
> Thank you for the thoughtful question. You are correct that $K$ is a set of learnable keys, and $W_k$ serves as a linear projection applied to them. We retain this projection mainly to follow standard practice in conventional attention mechanisms, where the separate linear layers allow greater representational flexibility and facilitate alignment with queries in the attention computation. We will add this clarification to our paper.
>
> > **Q: The authors mentioned that many key-value pairs are not heavily utilized. Do you also notice similar behavior with experts? Are only a few experts “activated” during inference, or is the coordinate-conditioning effective in activating them?**
>
> **A:**
> Thank you for this thoughtful question. We have **added a new section on expert utilization  (Appendix I)** to analyze this behavior. As shown in these additional results, **all experts are activated during inference**, and the coordinate-conditioned gating is effective. While some uneven usage occurs, it naturally reflects the non-uniformity of the scientific simulation data.
>
> > **Q: In this paper, I assume that the simulation parameters $p$ are finite-dimensional. Is it possible to also apply this method (with minor modifications) in the operator learning case when $p$ is an infinite-dimensional function?**
>
> **A:**
> Thank you for the insightful question. You are correct that our current formulation assumes that the simulation parameters $p$ are finite-dimensional. Extending FA-INR to an operator-learning setting, where $p$ may be an infinite-dimensional function (e.g., initial conditions or boundary conditions), is conceptually feasible.
>
> In such settings, the functional input $p(x)$ is typically discretized over the spatial domain. This discretized function will then be passed through an encoder network (similar to the branch network in DeepONet) to map it to a latent representation. This latent representation directly replaces the parameter embedding $z^{p_j}$ in our original FA-INR framework (now Line 289). While this extension is beyond the scope of our current work, the FA-INR architecture can be adapted to the operator learning setting with small modifications.

---

> > ### Author Response · Authors · 2025-12-03
> > **Reply to Reviewer jjGs: about revision**
> >
> > We appreciate the reviewer’s insightful comments again. We have **revised our Section 4.1 to clarify the purpose of $W_k$** in the cross-attention mechanism. Furthermore, we **updated our Figure 1** to include the corresponding **ground-truth visualization.**

---

### Official Review · Reviewer_xowU · 2025-10-30

**Soundness:** 3
**Presentation:** 3
**Contribution:** 4
**Rating:** 6
**Confidence:** 2

**Summary:**

This paper introduces Feature-Adaptive Implicit Neural Representation (FA-INR), a framework for building compact, high-fidelity surrogate models for scientific simulations. The INR uses explicit geometric structures increase memory cost and reduce adaptability. FA-INR replaces fixed grid embeddings with a learnable cross-attention–based memory bank for adaptive feature allocation, integrates a Mixture-of-Experts (MoE) routing mechanism based on spatial coordinates to improve scalability and specialization. The proposed method achieves up to 17% higher PSNR with 5–10× fewer parameters.

**Strengths:**

The paper proposes a novel method incorporating learnable attention memories that generalizes previous grid-based augmentations and introduces MoE specialization tailored to scientific fields with localized complexity.

The method is well-motivated, the architecture is clearly depicted. Visuals results convincingly illustrate qualitative gains.

**Weaknesses:**

FA-INR requires longer training due to attention and MoE operations, which may limit adoption in very large simulation campaigns.

Percentage gains in PSNR (dB) (e.g., “17.49% improvement” line 404) is not meaningful; absolute dB differences or linear-scale are more interpretable.

**Questions:**

Would you consider adding one operator-learning baseline (FNO with hypernetwork for parameters) to demonstrate FA-INR’s superiority beyond INR family methods.

---

> ### Author Response · Authors · 2025-11-23
> **Reply to Reviewer xowU**
>
> We sincerely appreciate the reviewer’s time and effort in reviewing our paper. We are glad that you found our method “novel” and “well-motivated,” and the model architecture is “clearly depicted”. We would like to address each of your concerns in detail below. We have already updated our paper with additional results, and the remaining revisions will be completed before December 3.
>
> > **Q: FA-INR requires longer training due to attention and MoE operations, which may limit adoption in very large simulation campaigns.**
>
> **A:**
> We thank the reviewer for highlighting this important consideration. We acknowledge that FA-INR requires slightly longer training time compared to methods that rely on direct interpolation from rigid explicit structures (e.g., K-Planes, Explorable-INR). However, *we respectfully believe this does not limit adoption in large simulation campaigns*.
>
> *First*, **the additional training time is negligible compared to the underlying simulator cost**. For example, Nyx requires 140 hours and MPAS-Ocean requires 83 hours to run the full-scale simulations. For the scenarios where hundreds or thousands of simulations may be needed, the one-time training cost is minimal compared to the substantial time saved through surrogate inference.
> *Second*, this extra computation from attention and MoE directly **leads to substantially higher fidelity achieved by FA-INR**. As shown in our experiments, FA-INR achieves state-of-the-art performance compared to the methods that rely on direct interpolation from rigid explicit structures (e.g., K-Planes, Explorable-INR). For scientific applications, this gain in accuracy is very crucial.
> *Third*, FA-INR remains extremely compact (e.g., 4.19 MB vs. 164.68 MB for K-Planes), making it suitable for large-scale deployment.
>
> Finally, the training cost can be further reduced through distributed computation, as both attention and MoE operations can be parallelized, and can also be enhanced through more efficient attention mechanisms. To reflect these points, we will include more discussions in the revised version of our paper.
>
>
> > **Q: Percentage gains in PSNR (dB) (e.g., “17.49% improvement” line 404) is not meaningful; absolute dB differences or linear-scale are more interpretable.**
>
> **A:**
> Thank you for the suggestion. To make the comparison more interpretable, we **have updated Sections 5.2 and 5.3** to report **the average absolute PSNR improvements**.
>
> > **Q: Would you consider adding one operator-learning baseline (FNO with hypernetwork for parameters) to demonstrate FA-INR’s superiority beyond INR family methods?**
>
> **A:**
> Thank you for the helpful suggestion. We tried our best to answer the question based on our knowledge, and we are open to further discussions. **We have incorporated a parameter-conditioned operator-learning baseline using a 3D Fourier Neural Operator with a hypernetwork (Hyper3D-FNO) based on the papers [1] [2].** This model takes the initial condition as input and uses a hypernetwork to generate FiLM (Feature-wise Linear Modulation) coefficients that adapt each FNO layer based on the simulation parameters. This design is conceptually aligned with previous hypernetwork-based operator-learning works such as HyperDeepONet [1] and FiLM-based modulation mechanisms used in HyPINO [2].
>
> We have the experimental results on the Nyx and CloverLeaf3D datasets. Due to memory constraints, the Nyx dataset was downsampled to $256^3$. *Hyper3D-FNO, under a similar model size to FA-INR, achieved a mean test PSNR of 42.93 (MD=0.0920) on CloverLeaf3D and 40.53 PSNR (MD=0.0908) on the downsampled Nyx dataset*. In contrast, our FA-INR achieves 53.40 PSNR on CloverLeaf3D and 44.70 PSNR on Nyx. We will include this additional baseline in our revised manuscript.
>
> [1] HyperDeepONet: Learning Operator with Complex Target Function Space using the Limited Resources via Hypernetwork, ICLR 2023.
>
> [2] HyPINO: Multi-Physics Neural Operators via HyperPINNs and the Method of Manufactured Solutions, NeurIPS 2025.

---

> > ### Author Response · Authors · 2025-12-03
> > **Reply to Reviewer xowU: about revision**
> >
> > We appreciate the reviewer’s insightful comments again. We have **revised the discussion in Appendix H** (in the updated paper) to provide additional clarification regarding the training time, and all revisions are highlighted in blue in the updated manuscript.

---

### Official Review · Reviewer_Mqe9 · 2025-11-01

**Soundness:** 2
**Presentation:** 3
**Contribution:** 2
**Rating:** 4
**Confidence:** 4

**Summary:**

This paper proposes Feature-Adaptive Implicit Neural Representations (FA-INR), an adaptive surrogate modeling framework for scientific simulations. FA-INR augments standard INRs with a cross-attention-based memory bank and an (Mixture-of-Experts) MOE module to improve fidelity and scalability in modeling complex physical fields. The paper evaluates the method on three benchmark datasets, such as MPAS-Ocean, Nyx, and CloverLeaf3D, showing quantitative gains in accuracy over existing INR-based surrogates.

**Strengths:**

- The topic of scientific surrogate modeling with neural fields is well aligned with the community’s growing interest in implicit neural representations and scientific machine learning.

- The work provides broad empirical validation across three simulation domains, along with targeted ablation studies on key architectural choices.

- The proposed method achieves better PSNR and SSIM metrics than INR-based baselines while maintaining a smaller model size.

**Weaknesses:**

- **Incremental technical novelty.** The paper primarily combines existing components, such as cross-attention mechanisms and Mixture-of-Experts (MoE), within a conventional INR framework, resulting in limited methodological innovation. Notably, [1] also employs cross-attention and MoE, where the query points can similarly be interpreted as coordinate inputs to an INR. The authors should clearly articulate how their proposed approach differs from [1] and from other related transformer-based operator-learning methods. Including an experimental comparison with such models would further strengthen the paper’s contribution.

- **Overstated claims of compactness and scalability.** Although FA-INR employs fewer parameters than some baselines, the paper itself acknowledges increased training time due to the use of attention and MoE modules. Therefore, the claimed efficiency advantages appear overstated.

- **Lack of diverse baselines and discussion of related work.** The experimental evaluation is confined to INR-based models, lacking comparisons with broader classes of surrogate modeling approaches. In particular, neural operator methods, commonly used for scientific surrogate modeling and sharing a similar problem formulation where simulation parameters serve as inputs, are neither discussed nor compared. Furthermore, the paper omits discussion of other implicit neural representation methods applied to scientific forecasting [2, 3]. Including these perspectives would provide a more balanced and comprehensive assessment of the proposed method’s performance and relevance.

[1] GNOT: A General Neural Operator Transformer for Operator Learning, ICML, 2023.

[2] Continuous PDE Dynamics Forecasting with Implicit Neural Representations, ICLR, 2023.

[3] CROM: Continuous Reduced-Order Modeling of PDEs Using Implicit Neural Representations, ICLR, 2023.

**Questions:**

See weaknesses.

---

> ### Author Response · Authors · 2025-11-23
> **Reply to Reviewer Mqe9 (1/2)**
>
> We sincerely appreciate the reviewer’s time and effort in reviewing our paper. We are glad that you found the topic “well aligned with the community’s growing interest.” Below, we provide detailed responses to the concerns you raised. We have already updated our paper with additional results, and the remaining revisions will be completed before December 3.
>
> > **Q: Incremental technical novelty. The paper primarily combines existing components, such as cross-attention mechanisms and Mixture-of-Experts (MoE), within a conventional INR framework, resulting in limited methodological innovation.
> Notably, [1] also employs cross-attention and MoE, ... The authors should clearly articulate how their proposed approach differs from [1] and from other related transformer-based operator-learning methods.**
>
> **A:**
> We thank the reviewer for this thoughtful comment. Although FA-INR leverages established components such as cross-attention and Mixture-of-Experts (MoE), the integration is far from trivial. We note that our main goal is to improve INR for the scientific simulation surrogate, and we argue that 1) how to make the model adaptive to data characteristics, and 2) how to make the model scalable to large-scale simulation data, are the main bottlenecks. **Our contribution lies in identifying these weaknesses and purposefully integrating and adapting these mechanisms to address specific limitations in the existing INR-based surrogate models**, rather than simply reusing these mechanisms (Lines 88-101). In our humble opinion, such a process and proposed solution are *by no means trivial*. **Repurposing existing techniques to address new problems should not be seen as a weakness**.
>
> We also appreciate the reviewer pointing out the paper [1], which we have cited in our revised manuscript. We have **added a discussion of transformer-based operator-learning approaches, including GNOT [1], to the related work section** (now Lines 145-157).
> After a careful review, we clarify that, while both FA-INR and GNOT [1] incorporate cross-attention and Mixture-of-Experts (MoE) components, **they are designed for fundamentally different problems and different modeling assumptions**.
>
> GNOT is a neural operator framework that learns mappings between function spaces, with a focus on handling heterogeneous input functions and irregular meshes, which are the key challenges in operator learning. In contrast, our FA-INR is an implicit neural representation (INR) model designed to directly map input coordinates and simulation parameters to physical values. Our focus is on high-fidelity field modeling for **high-resolution 3D** ensemble simulations, including both **large-scale volumetric** data and **irregular mesh** data. In our setting, the spatial domain is dense and the initial condition is fixed, and the primary goal is accurate surrogate modeling, which is an essential task for **analyzing how simulation parameters influence physical outputs, rather than operator generalization**.
>
> **The role of cross-attention is also fundamentally different**. In FA-INR, cross-attention is used to retrieve features from a static learnable key-value memory bank given the coordinate queries. This design enables adaptive spatial encoding and efficient feature reuse in large volumetric domains. In GNOT, the Heterogeneous Normalized Cross-Attention (HNA) is used to aggregate embeddings of multiple input functions and meshes. The attention enables heterogeneous input fusion, which is not applicable in our setting.
>
> **The MoE mechanisms serve different purposes**. GNOT applies a conventional FFN-based MoE structure within Transformer layers, and MoE is applied only after the cross-attention and self-attention mechanisms. In contrast, FA-INR uses MoE to route spatial coordinates to specialized memory banks within its cross-attention mechanism, enabling spatially adaptive feature retrieval rather than FFN specialization.
>
> Finally, regarding scalability for high-resolution 3D ensemble simulations, **applying GNOT to the large-scale volumetric datasets presents significant computational challenges**. For example, a single Nyx field at $512^3$ resolution involves 134 million input points. Processing such sequences with Transformer-based neural operators is infeasible on standard GPU setups due to memory constraints. In contrast, FA-INR processes the coordinate inputs individually, allowing efficient training and inference on large-scale 3D fields.
>
> That said, we are **actively evaluating** the feasibility of adapting GNOT to our task on MPAS-Ocean and CloverLeaf3D datasets. Our preliminary investigation shows that there are non-trivial modifications to make, and we will update our findings or results once we have them.

---

> > ### Author Response · Authors · 2025-11-23
> > **Reply to Reviewer Mqe9  (2/2)**
> >
> > > **Q: Overstated claims of compactness and scalability. Although FA-INR employs fewer parameters than some baselines, the paper itself acknowledges increased training time due to the use of attention and MoE modules. Therefore, the claimed efficiency advantages appear overstated.**
> >
> > **A:**
> > Thank you for pointing this out. We apologize for any confusion. We want to clarify this potential misunderstanding.  Our claims of **“efficiency” primarily refer to model compactness and representational scalability, not training time**. While FA-INR introduces additional training cost due to attention and MoE operations, we humbly think this is acceptable, *as analogous to model-pruning pipelines*, which achieve reduced model sizes at the expense of iterative, longer training. We will revise our paper to avoid any misunderstanding.
> >
> >
> > > **Q: Lack of diverse baselines and discussion of related work. The experimental evaluation is confined to INR-based models, lacking comparisons with broader classes of surrogate modeling approaches.
> > In particular, neural operator methods, commonly used for scientific surrogate modeling and sharing a similar problem formulation where simulation parameters serve as inputs, are neither discussed nor compared.
> > Furthermore, the paper omits discussion of other implicit neural representation methods applied to scientific forecasting [2, 3]. Including these perspectives would provide a more balanced and comprehensive assessment of the proposed method’s performance and relevance.**
> >
> > **A:**
> > We appreciate this valuable comment. Regarding baselines, we want to respectfully emphasize that our paper's primary objective has never been to develop the optimal surrogate modeling approach in a broad sense. Rather, our work is focused specifically on advancing Implicit Neural Representations (INRs) for high-fidelity scientific simulation surrogates on large-scale 3D ensemble datasets. *The advantages of INRs for surrogate modeling have been extensively investigated in prior studies (now Lines 158-165), and we compare our approach against the most suitable and state-of-the-art INR-based methods mainly to demonstrate and verify our contributions (now Lines 168-173)*.
> >
> > With that being said, we appreciate the reviewer bringing up this point and will revise our related work section to include the discussion. At a high level, neural operator methods (NOs) could be adapted to address our problem, but **the data, assumptions, and problem formulation differ fundamentally**. Our setting assumes the *fixed initial conditions* and *varying finite-dimensional simulation parameters*. In contrast, NOs take function inputs, such as initial-condition fields or boundary-condition functions, and learn mappings between function spaces.
> >
> >
> > Therefore, **the role and representation of simulation parameters are fundamentally different**: in NOs, parameters are naturally embedded as spatially distributed functions; whereas in our problem, they appear as global parameter vectors that must influence every coordinate query. This makes **parameter conditioning much more challenging in our setting** due to the highly nonlinear relationship between parameters and target fields. While one could adapt NO architectures for our problem, doing so would require substantial structural changes. In our humble opinion, making the NO-based frameworks work well on our task may deserve another paper.
> >
> > Furthermore, we *thank the reviewer for pointing out the INR-based approaches, such as [2][3], in the context of scientific forecasting*. After carefully reviewing these works, we agree that they demonstrate the broader applicability of implicit neural representations (INRs) in modeling scientific tasks. Specifically, both works adopt INRs in a domain-specific way, driven by the uniqueness of their respective PDE modeling tasks. For example, DINo [2] modified INR for ODE-based forecasting with space-time separation. CROM [3] adapts INRs into discretization-independent manifolds for reduced-order modeling. These also reflect our design philosophy: we leverage INRs’ flexibility while introducing architectural improvements for the specific challenges of our surrogate modeling task. We will include the discussion of INR-related works for scientific forecasting in our revised paper.
> >
> > [1] GNOT: A General Neural Operator Transformer for Operator Learning, ICML, 2023.
> >
> > [2] Continuous PDE Dynamics Forecasting with Implicit Neural Representations, ICLR, 2023.
> >
> > [3] CROM: Continuous Reduced-Order Modeling of PDEs Using Implicit Neural Representations, ICLR, 2023.

---

> > > ### Author Response · Authors · 2025-12-03
> > >
> > > We appreciate the reviewer’s insightful comments again. Here is our additional response.
> > >
> > > 1. We follow your suggestion to evaluate **neural-operator-based methods,** specifically the **GNOT algorithm.** On the 3D simulation datasets used in our experiments, we confirm **our concern about GNOT regarding its computational and memory limitations,** as mentioned in our response in **Reply to Reviewer Mqe9 (1/2).** Because GNOT processes the entire field as a sequence, it encounters substantial memory constraints even on our *smallest* dataset (i.e., CloverLeaf3D with $128^3$ resolution). As a result, we conducted the evaluation on a downsampled version of CloverLeaf3D ($64^3$). On this setting, GNOT (1.25M parameters) achieved 39.37 PSNR (MD = 0.0899), whereas our FA-INR (1.10M parameters) achieved 52.45 PSNR (MD = 0.0458) on the same downsampled dataset. **These results further highlight the challenges of directly applying neural operator learning methods to the high-resolution 3D ensemble simulation datasets.**
> > >
> > > 2. In addition, we have **revised the related work section** to include discussions of both transformer-based operator-learning methods and INR-based works for scientific forecasting.
> > >
> > > Once again, we appreciate your comments, and we believe we have addressed most, if not all, of them.

---

### Meta-Review · Area_Chair_N9DX · 2026-01-06

**Summary:**

This paper proposes a Feature-Adaptive implicit neural representation approach for surrogate modeling. The method leverages cross-attention and incorporates a coordinate-guided mixture-of-experts (MoE) mechanism. While the approach is implemented carefully and the empirical results are promising, reviewers note that the method largely combines existing components in a new configuration rather than introducing fundamentally new ideas.

A central concern is that the paper provides limited justification for the design choices: no theoretical analysis is offered, leaving it unclear when and why the proposed architecture should outperform the baselines. Given the empirical nature of the work, the evaluation would benefit from a broader and more contextualized set of comparisons. In particular, comparisons to relevant physics-based methods are missing, and the paper’s claims of substantially faster performance are not supported by a clear accuracy–computation trade-off analysis. Additionally, the ablation study is difficult to interpret, and it remains unclear which components contribute most to the observed gains.

The authors used the rebuttal phase to address several points raised by the reviewers, but concerns remain regarding novelty, clarity, and positioning. In particular, it is still not clear how this work advances machine learning beyond applying a combination of known techniques to a specific surrogate modeling setting, nor does it provide new insights or understanding of the underlying problem. Overall, the motivation for the particular architecture remains somewhat ad hoc, and the revisions do not fully resolve the reviewers’ core concerns.

In light of the remaining issues and the mixed reviews, I am leaning toward reject in its current form. This said, I encourage to resubmit a revised version to another ML venue.

**Reviewer Concerns:**

The authors provided additional experiments and incorporated additional baselines during the rebuttal. However, it remains unclear whether the evaluation is sufficient for a empirical paper. In particular, key questions regarding technical novelty and the broader insights offered by the work remain largely unresolved. Moreover, the value of implicit neural representations (INRs) as a paradigm for accelerating scientific simulations is not fully justified. The paper largely assumes that INRs are a good and effective starting point, and focuses on improving within this framework; however, this assumption itself is debatable and would benefit from stronger motivation and contextualization.

**Reviewer Scores:**

I don't believe the discussion phase would have shifted the scores much.

---

### Decision · Program_Chairs · 2026-01-26

Reject